# Collective Counterfactual Explanations: Balancing Individual Goals and Collective Dynamics

**Ahmad-Reza Ehyaei**
Max Planck Institute for Intelligent Systems, Tübingen AI Center, Tübingen, Germany
ahmad.ehyaei@tuebingen.mpg.de

**Ali Shirali**
University of California, Berkeley, USA
shirali_ali@berkeley.edu

**Samira Samadi**
Max Planck Institute for Intelligent Systems, Tübingen AI Center, Tübingen, Germany
ssamadi@tuebingen.mpg.de

## Abstract

Counterfactual explanations provide individuals with cost-optimal recommendations to achieve their desired outcomes. However, when a significant number of individuals seek similar state modifications, this individual-centric approach can inadvertently create competition and introduce unforeseen costs. Additionally, disregarding the underlying data distribution may lead to recommendations that individuals perceive as unusual or impractical. To address these challenges, we propose a novel framework that extends standard counterfactual explanations by incorporating a population dynamics model. This framework penalizes deviations from equilibrium after individuals follow the recommendations, effectively mitigating externalities caused by correlated changes across the population. By balancing individual modification costs with their impact on others, our method ensures more equitable and efficient outcomes. We show how this approach reframes the counterfactual explanation problem from an individual-centric task to a collective optimization problem. Augmenting our theoretical insights, we design and implement scalable algorithms for computing collective counterfactuals, showcasing their effectiveness and advantages over existing recourse methods, particularly in aligning with collective objectives.

## 1 Introduction

Algorithmic decisions are increasingly shaping various aspects of our lives, including our access to opportunities and services Karimi et al. [2022]. For individuals negatively affected by an algorithmic decision (e.g., loan denial), counterfactual explanations (**CE**) Wachter et al. [2017] provide actionable insights by identifying minimal changes (e.g., increasing savings) needed to achieve a favorable outcome (e.g., loan approval). Defining CE requires three elements:

1. A feature space $\mathcal{X} \subseteq \mathbb{R}^d$ characterizes individuals and is equipped with a probability measure $\mathbb{P} \in \mathcal{P}(\mathbb{R}^d)$, where $\mathcal{P}(\mathbb{R}^d)$ represents the set of all probability measures on $\mathbb{R}^d$.

2. A cost function $c : \mathcal{X} \times \mathcal{X} \to \mathbb{R}$ quantifies the effort to modify features $x \in \mathcal{X}$ to $x' \in \mathcal{X}$.

39th Conference on Neural Information Processing Systems (NeurIPS 2025).

3. A binary classifier $h : \mathcal{X} \to \mathcal{Y} \in \{\pm 1\}$ assigns a decision to every $x \in \mathcal{X}$. This partitions $\mathcal{X}$ into undesirable $\mathcal{X}^- = \{x \in \mathcal{X} : h(x) = -1\}$ and desirable $\mathcal{X}^+ = \{x \in \mathcal{X} : h(x) = +1\}$ subsets. The respective probability distributions, $\mathbb{P}_-$ and $\mathbb{P}_+$, are induced by restricting $\mathbb{P}$ to $\mathcal{X}^-$ and $\mathcal{X}^+$.

For an individual $x \in \mathcal{X}^-$ receiving an undesirable outcome, the counterfactual explanation $\mathbf{CE}(x)$ provides the minimal-cost strategy to achieve the favorable label:

$$\mathbf{CE}(x) = \underset{x' \in \mathcal{X}^+}{\arg\min} \left\{ c(x, x') \right\}. \tag{1}$$

The standard CE formulation assumes that individuals act in isolation, ignoring interactions between individuals moving to the same destination in $\mathcal{X}^+$. However, real-world dynamics are far more complex. While an individual benefits from transitioning from $\mathcal{X}^-$ to $\mathcal{X}^+$ by receiving a positive decision, other factors can influence their overall utility. For instance, individuals moving to an overcrowded region may face increased competition, leading to a decline in their utility. By solving CE on a per-individual basis, the standard formulation overlooks these broader societal impacts, failing to account for the collective consequences of explanations. This so-called *externality* resembles the *tragedy of the commons* Gross and De Dreu [2019], as seen in navigation algorithms optimizing routes independently.

In our work, we model competition by assuming a fixed, yet unknown, amount of resources is available for individuals with a specific feature $x$. For instance, if $x$ represents job-relevant attributes, the resources at $x$ correspond to the societal demand for the expertise associated with $x$. While these resources are not directly observable, modeling population dynamics allows us to link them to the population distribution at equilibrium. Assuming the population is in equilibrium before CE generation, the current population density $\mathbb{P}_+$ serves as a strong predictor of the available resources.

Leveraging the connection between current population density and available resources, we propose a framework called *Collective Counterfactual Explanation* (**CCE**). CCE accounts for the limited resources at each $x$ and generates explanations collectively. It guides individuals in a way that the population, after receiving and partially following these explanations, reaches a state close to equilibrium. This ensures that the externalities from increased competition are minimized, making the generated explanations more reliable and beneficial for everyone.

Neglecting the underlying distribution $\mathbb{P}$ raises concerns about robustness to inaccurate cost estimates or feasibility constraints. Social structures and unobserved costs may have already pushed individuals out of low-density areas, rendering recommendations towards those regions ineffective. To address this, recent work highlights *data manifold closeness* as a key requirement in CE Karimi et al. [2022], Guidotti [2022], Verma et al. [2020]. As we discuss in Sec. 3, the CCE formulation naturally aligns with this and related desiderata.

To illustrate these nuances, we use the Moons dataset Pedregosa et al. [2011] with a non-linear SVM classifier and decision boundary $L$ in Fig. 1. Standard CE methods, such as Wachter et al. [2017], guide all applicants to the decision boundary $L$. While this approach is cost-effective, it neglects the feature-space distribution $\mathbb{P}$, potentially concentrating individuals in a low-resource region (left panel). This issue is particularly pronounced for SVM classifiers, which emphasize the margin between $L$ and the nearest data points. In contrast, CCE generates a more natural distribution, balancing the costs incurred by individuals with their impact on others (right panel).

**Our Contributions.** We leverage a population dynamics model from mean-field game theory to incorporate competitive interactions between individuals into the CE formulation. Our framework, Collective Counterfactual Explanations (CCE), penalizes deviations from equilibrium to minimize unnecessary competition costs pursuant to the following recommendations. We propose a relaxed version of CCE that reformulates the CE generation problem as an unbalanced optimal transport (OT) problem. This reformulation enables us to draw on extensive techniques and tools from the OT literature to address various challenges in CE. In sum, our main contributions are:

- Formalize the collective costs of CE using a model of population dynamics (Sec. 2.1).
- Propose the CCE framework that penalizes externalities from individual interactions (Sec. 2.2).
- Relax CCE to unbalanced OT and provide solution existence and consistency guarantees (Sec. 2.3).
- Design an efficient algorithm to solve CCE with the benefit of amortized inference (Sec. 2.4).
- Demonstrate the advantage of CCE over standard CE along various desiderata (Sec. 3).

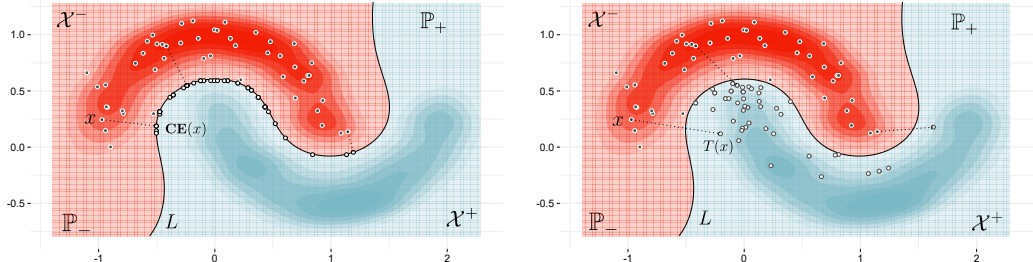

Figure 1: Comparing CE methods with a non-linear SVM classifier. (Left) Wachter et al. [2017] place all recommendations on the decision boundary $L$. (Right) In contrast, Collective CE moves individuals to more populated areas, which are potentially more resource-rich.

- Extend CCE to a wide range of new settings and problems, including temporal recourse and recourse based on an ordered family of classifiers (Sec. 4).

- Conduct numerical studies to support the theoretical results and the efficacy of our method (Sec. 5).

## 2 Collective Counterfactual Explanation (CCE)

In this section, we extend the standard CE framework to address resource scarcity and competition arising from CE. The standard framework assumes individuals seek the minimum-cost action to reach $\mathcal{X}^+$. Therefore, it considers the cost of change and the benefit of positive classification as the only factors important for the individual. However, in practice, the benefits of positive classification are not uniform; when individuals cluster at specific points in $\mathcal{X}^+$, congestion can reduce these benefits.

We model congestion and reduced benefits by assigning a resource to each $x \in \mathcal{X}$ and denote the *resource distribution* by $\mathbb{S} \in \mathcal{P}(\mathcal{X})$. If we ignore resource limitations, following CE may increase demand beyond the resources available at some points. This inefficiency can lower overall utility or force individuals to take further actions to establish a new equilibrium, which may undermine the value of our recommendations over time.

An ideal collective CE should transport the population from one equilibrium to another. We assume that individuals are in equilibrium before CE. Our goal is to generate CE that moves individuals to a new equilibrium, under the assumption that a random subset of individuals fully comply with the recommendations. Suppose there exists a function $E : \mathcal{P}(\mathcal{X}) \to \mathbb{R}$ that can measure the distance of a distribution from equilibrium. Denote the distribution over $\mathcal{X}^+$ after CE by $\mathbb{P}_{\mathbf{CE}}$. Then, to ensure the recommendations lead to a new equilibrium, we can include a penalty $E(\mathbb{P}_{\mathbf{CE}})$ in the objective.

A key property of equilibrium enables us to design a function $E$ that penalizes deviations from it. While the resources at each $x$ are not directly observable, mean-field game theory links the resource and equilibrium distributions. Specifically, in Sec. 2.1, we show that $\mathbb{P} \propto \mathbb{S}$ under equilibrium. Consequently, if CE results in an equilibrium, it should satisfy $\mathbb{P}_{\mathbf{CE}} \propto \mathbb{S}_+$, where $\mathbb{S}_+$ represents the resource distribution over $\mathcal{X}^+$. To quantify deviations from equilibrium, we measure how far $\frac{d\mathbb{P}_{\mathbf{CE}}}{d\mathbb{P}_+}$ (known as Radon-Nikodym derivative) is from 1. In particular, we will define $E(\mathbb{P}_{\mathbf{CE}})$ is as the $\chi^2$-divergence between $\mathbb{P}_{\mathbf{CE}}$ and $\mathbb{P}_+$ which can capture the extent of deviation effectively.

Under mild assumptions (see Lemma 1), we can express CE as a *mapping* $T$ from $\mathcal{X}^-$ to $\mathcal{X}^+$. Let $\mathcal{M}(\mathcal{X}^-, \mathcal{X}^+)$ denote the space of all such mappings. To describe the distribution of individuals who were initially negatively classified and follow the CE, we use $T_\#\mathbb{P}_-$, the push-forward distribution by the map $T$. With this terminology, the collective CE problem solves the following problem:

**Proposition 1 (Collective Counterfactual Explanation)** *Under the population dynamics described in Sec. 2.1, and assuming a $\gamma$ proportion of individuals in $\mathcal{X}^-$ follow the explanations, CCE solves*

$$\arg\min_{T \in \mathcal{M}(\mathcal{X}^-, \mathcal{X}^+)} \left\{ \mathop{\mathbb{E}}_{x \sim \mathbb{P}_-} [c(x, T(x))^q]^{\frac{1}{q}} + \eta \lambda^2 \, \mathrm{D}_{\chi^2} \left( T_\# \mathbb{P}_- \,\|\, \mathbb{P}_+ \right) \right\}, \tag{2}$$

*for $q \in [1, \infty)$ and a competition regularization $\eta$. Here, $\lambda = \dfrac{\gamma p_-}{\gamma p_- + p_+}$, where $p_+$ and $p_-$ are the proportions of the population in $\mathcal{X}^+$ and $\mathcal{X}^-$, respectively.*

The solution to Eq. (2) may not always exist Royden and Fitzpatrick [2010]. To address this, we introduce a relaxed version of CCE with existence and consistency guarantees in Sec. 2.3.

The rest of this section outlines the tools and details to derive CCE in Prop. 1. We begin by introducing population dynamics and equilibrium (Sec. 2.1). Then, we provide a step-by-step derivation of CCE (Sec. 2.2), and its relaxed version (Sec. 2.3). We conclude with algorithms to solve CCE (Sec. 2.4).

## 2.1 Population Dynamics and Equilibrium

Let $\mathbb{U}(\cdot, t) \in \mathcal{P}(\mathcal{X})$ be the distribution of individuals over $\mathcal{X}$ at time $t$. To analyze $\mathbb{U}$, we use a mean-field game theoretic framework Lasry and Lions [2006], Carmona et al. [2018]. In this framework, the utility of an individual at $x$ depends on the density of resources $\mathbb{S}(x)$ as well as competition which arises from the local population density $\mathbb{U}(x, t)$. This interplay between the attraction of resources and the pressure of competition drives how the population redistributes itself over time. Mathematically, the following PDE explains the evolution of the population:

$$\frac{\partial \mathbb{U}(x, t)}{\partial t} = \nabla \cdot \Big( \mathbb{U}(x, t) \, \nabla \big( \beta \mathbb{U}(x, t) - \alpha \mathbb{S}(x) \big) \Big). \tag{3}$$

- Attraction: $\nabla \mathbb{S}$ captures how individuals move in response to resource availabilities.
- Competition: $\nabla \mathbb{U}$ captures how individuals spread out depending on the presence of each other.
- Together, $\mathbf{g} := \nabla \left( \alpha \mathbb{S}(x) - \beta \mathbb{U}(x, t) \right)$ is the driving gradient. Individuals move along this gradient. The parameters $\alpha > 0$ and $\beta > 0$ determine the significance of resource attraction and competition.
- The term $\mathbb{U} \mathbf{g}$ is the flux of the population, which is proportional to both $\mathbb{U}$ and the driving gradient.

At equilibrium, the population equilibrium density $\mathbb{U}^*(x)$ satisfies $\nabla \big( \beta \mathbb{U}^*(x) - \alpha \mathbb{S}(x) \big) = 0$ which implies $\mathbb{U}^*(x) = \frac{\alpha}{\beta} \mathbb{S}(x) + C$, for some constant $C$ (refer to Appendix C for additional details).

## 2.2 Formal Derivation of CCE Formulation

In the first step, we reformulate the standard CE problem in Eq. (1) as an optimization over the space of measurable functions. Let $\mathcal{M}(\mathcal{X}^-, \mathcal{X}^+)$ denote the space of all measurable functions mapping from the subspace $\mathcal{X}^-$ to $\mathcal{X}^+$. Using Lemma 1 (which we deferred to the appendix for brevity), we reformulate CE as minimizing the cost function within the space of measurable maps on $\mathcal{M}(\mathcal{X}^-, \mathcal{X}^+)$. Formally, for any $q \in [1, \infty)$, the CE in Eq. (1) is equivalent to

$$\arg\min_{T \in \mathcal{M}(\mathcal{X}^-, \mathcal{X}^+)} \left\{ \mathop{\mathbb{E}}_{x \sim \mathbb{P}} [c(x, T(x))^q]^{\frac{1}{q}} \right\}. \tag{4}$$

The second step is to incorporate an additional term in the objective to penalize deviations from equilibrium. As shown in Sec. 2.1, equilibrium requires the distribution of individual features to align with the resource distribution. Assuming the population starts at equilibrium and the resource distribution remains unchanged after intervention, we can measure deviations from equilibrium by comparing $\mathbb{P}_+$ and $\mathbb{P}_{\mathbf{CE}}$, i.e., the distribution over $\mathcal{X}^+$ before and after CE. A general measure of $\varphi$-divergence can quantify this difference. We particularly use $\chi^2$-divergence corresponding to $\varphi(t) = (t - 1)^2$ as this will make the connection between CCE and optimal transport theory explicit.

The third and final element needed to derive CCE in Prop. 1 is a model of individual responses to CE. We assume that a $\gamma$ proportion of individuals in $\mathcal{X}^-$ follow the CE to transition to $\mathcal{X}^+$. Let $p_+$ and $p_-$ represent the fractions of the population initially in $\mathcal{X}^+$ and $\mathcal{X}^-$, respectively. Under this response model, the distribution of individuals in $\mathcal{X}^+$ after receiving CE is $\mathbb{P}_{\mathbf{CE}} = \lambda T_\# \mathbb{P}_- + (1 - \lambda) \mathbb{P}_+$. This completes the preliminaries to prove Prop. 1, and we leave other details to the appendix.

## 2.3 Relaxation of CCE

Generally, there is no guarantee for the existence of a solution to the CCE problem in Eq. (2). A common technique to get around this is to search for an optimal *plan* $\pi \in \mathcal{P}(\mathcal{X}^- \times \mathcal{X}^+)$ instead of an optimal map. We refer the reader to Appendix B for additional context on optimal transport (OT) theory. Using this technique, we relax Eq. (2) with

$$\underset{\pi \in \mathcal{P}(\mathcal{X}^- \times \mathcal{X}^+)}{\arg \min} \left\{ \underset{(x,y) \sim \pi}{\mathbb{E}} \left[ c(x,y)^q \right]^{\frac{1}{q}} + \eta \lambda^2 \, \mathrm{D}_{\chi^2}\left(\pi_2 \,\|\, \mathbb{P}_+\right) \quad \text{s.t.} \quad \pi_1 = \mathbb{P}_- \right\}, \tag{5}$$

Where $\pi_1$ and $\pi_2$ are two marginal densities corresponding to the first and second coordinates of $\pi$. We formally prove the existence of a solution for this relaxed problem:

**Proposition 2 (Existence of a Plan)** *When $\mathcal{X} \subseteq \mathbb{R}^d$, $c : \mathcal{X} \times \mathcal{X} \to [0, \infty)$ is a lower semicontinuous cost function, $\eta > 0$, and $q \geq 1$, the relaxed CCE in Eq. (5) has a solution $\pi_* \in \mathcal{P}(\mathcal{X}^- \times \mathcal{X}^+)$.*

To explicitly connect to OT and leverage its extensive tools, we introduce an additional relaxation to Eq. (5): We replace the hard constraint $\pi_1 = \mathbb{P}_-$ with a penalty term $\lambda_1 \, \mathrm{D}_\psi(\pi_1 \,\|\, \mathbb{P}_-)$ to define.

$$\underset{\pi \in \mathcal{P}(\mathcal{X}^- \times \mathcal{X}^+)}{\arg \inf} \left\{ \underset{(x,y) \sim \pi}{\mathbb{E}} \left[ c(x,y)^q \right]^{\frac{1}{q}} + \lambda_1 \, \mathrm{D}_\psi(\pi_1 \,\|\, \mathbb{P}_-) + \lambda_2 \, \mathrm{D}_{\chi^2}(\pi_2 \,\|\, \mathbb{P}_+) \right\}. \tag{6}$$

Here, $\lambda_2 := \eta \lambda^2$ and $\lambda_1$ are regularization parameters and the choice of $q$ and $\mathrm{D}_\psi$ are arbitrary. We show that this relaxed version is, in fact, consistent in the following sense:

**Proposition 3** *Denote the solution to Eq. (5) by $\pi_*$ and the solution to its relaxed problem Eq. (6) by $\pi_{\lambda_1}$. As $\lambda_1 \to \infty$, we have $\pi_{\lambda_1} \to \pi_*$.*

## 2.4 Algorithms for CCE

We can build on extensive algorithmic tools from OT to design algorithmic solutions for CCE. Choosing $\mathrm{D}_\psi = \mathrm{D}_{\mathrm{KL}}$ and $q = 1$ in Eq. (6), we present a projected-gradient method to solve CCE in Algorithm 1 in the appendix. This algorithm has the following time complexity:

**Proposition 4 (Complexity of Algorithm 1)** *Let $m$ and $n$ be the sizes of the discrete sets $\mathcal{X}^-$ and $\mathcal{X}^+$, respectively, and let $T$ be the number of iterations in the for-loop (lines 3–8) of Algorithm 1. Then, the overall time complexity of the gradient-based unbalanced optimal transport solver is $O(T\,m\,n)$.*

Resembling the well-known Sinkhorn algorithm Peyré et al. [2017], Séjourné et al. [2022], Pham et al. [2020], we also present Algorithm 4, a fast gradient-based unbalanced OT solver for relaxed CCE. We refer the reader to the appendix for further details.

# 3 Collective Counterfactual Explanation Aligns with Key Desiderata

In this section, we show that the CCE framework not only accounts for limited resources and competition but satisfies key desiderata for a successful recourse highlighted in recent surveys Verma et al. [2020], Karimi et al. [2022].

**Data Manifold Closeness.** To avoid outliers and ensure CE is credible, it is suggested that CE remain close to the current data distribution Hamer et al. [2023], Movin et al. [2024]. This property, known as *data manifold closeness*, preserves intrinsic feature correlations and leads to more realistic, actionable recommendations. In CCE, penalizing deviations from equilibrium naturally enforces similarity to the current distribution. CCE also allows control over the desired level of closeness to the data manifold.

**Security & Privacy.** Each CE reveals that instances within a radius $c(x, \mathbf{CE}(x))$ around $x$ belong to the negative class, making the decision boundary easy to identify. Thus, standard CE APIs pose security risks by enabling low-cost construction of surrogate models Pawelczyk et al. [2023], Pentyala et al. [2023], Yang et al. [2022], as shown in Fig. 2 (middle). In contrast, CCE issues collective recommendations that account for equilibrium shifts, integrating diverse factors that intuitively complicate boundary manipulation, as shown in Fig. 2 (right).

**Robustness & Individual Fairness.** In real-world decision-making, robustness and individual fairness require similar individuals to receive comparable recommendations Ehyaei et al. [2023b], Guyomard et al. [2023], Artelt et al. [2021]. Building on Otto et al. [2021], we can see that under mild conditions on the densities $\mathbb{P}_-$, $\mathbb{P}_+$, and a general class of costs, CCE defines a diffeomorphism. Since the feature space $\mathcal{X}$ is typically compact, small feature changes result in small changes in CCE recommendations. In contrast, standard CE may yield sharply different outputs near the decision boundary, leading to unequal treatment of similar individuals.

**Amortized Inference.** The standard CE formulation requires solving an optimization problem for each individual, which can be computationally expensive. Amortized inference addresses this challenge by leveraging patterns learned from prior instances Verma et al. [2021], De Toni et al. [2023], Majumdar and Valera [2024]. Our CCE formulation employs a map $T$ or plan $\pi$ to generate explanations. Leveraging the rich literature of OT, we can efficiently learn this map or plan using numerical methods. Once learned, we can generate CE for any instance without additional computation.

**Actionability.** Certain features, such as birthplace (an immutable attribute) or age (which follows a naturally increasing trajectory), introduce additional constraints on the design of CE. Actionable recourse requires that recommendations respect these constraints Ustun et al. [2019], Joshi et al. [2019], Rawal and Lakkaraju [2020]. A common approach to handle such constraints is to assign large penalties in the cost function. In CCE, alternatively, we encode these constraints as linear constraints Zaev [2015] within the OT problem. This is feasible because Boolean functions—limits of compact support continuous functions—can generally be formulated as linear constraints. The following proposition formally establishes the existence of a valid CCE plan, assuming at least one plan satisfies the specified linear constraints.

**Proposition 5 (Actionable CCE Through Linear Constraints)** *Under conditions of Prop. 2, the relaxed CCE problem with linear constraints has a solution if and only if the set $\Gamma_{\mathcal{W}} := \{\pi \in \mathcal{P}(\mathcal{X}^- \times \mathcal{X}^+) : \int w d\pi = 0, w \in \overline{\mathcal{W}}\}$ is not empty, where $\overline{\mathcal{W}}$ is the closure of $\mathcal{W}$, a subset of continuous functions with compact support on the space $\mathcal{X}^- \times \mathcal{X}^+$.*

## 4 Extensions of Collective Counterfactual Explanation

In Sec. 2, we reformulated CE as the problem of finding an optimal coupling between the distributions $\mathbb{P}_-$ and $\mathbb{P}_+$. This perspective offers a unified framework for addressing key challenges in CE that are otherwise difficult to resolve under standard formulations. We outline two such challenges below.

### 4.1 Path-Guided Counterfactual Explanation

Path-guided CE extends standard CE by offering not just a final target point, but a sequence of intermediate steps that guide an input toward a desired outcome. To achieve this, we can use displacement interpolation in dynamic OT, where mass moves continuously over time from a source distribution to a target distribution.

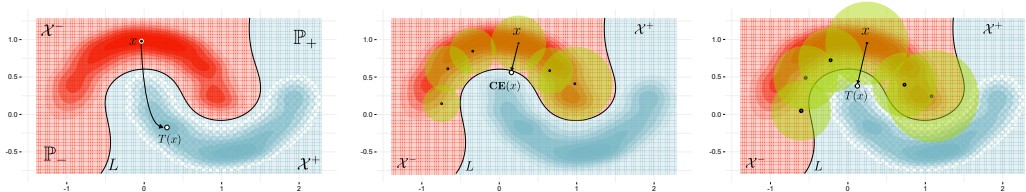

Figure 2: (Left) An example CCE recommendation. (Center) The standard CE method poses a higher risk of revealing the classifier boundary. (Right) Identification strategies are less effective in uncovering the boundary with the more sophisticated design CCE method.

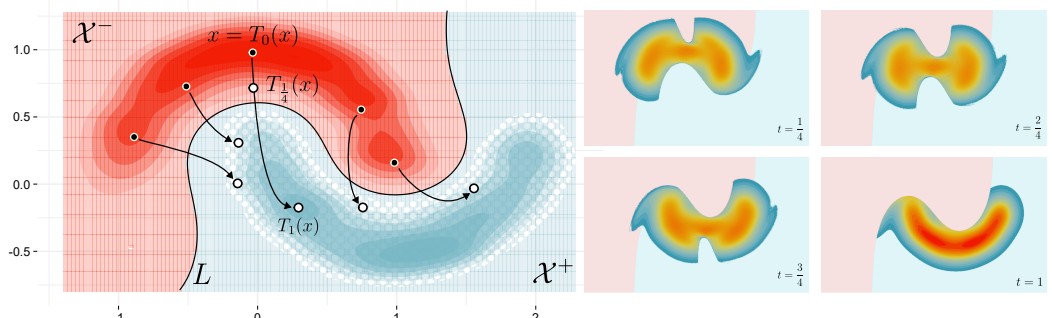

Figure 3: (Left) The temporal map showing the flow of each point as it moves toward the recourse target. (Right) The back-and-forth method was applied to estimate the CE map. By leveraging displacement interpolation, the optimal flow is depicted across four time steps, showing the transition of the negative region into the positive region using the Moons dataset.

To introduce a temporal dimension into CE, we define a time-indexed family of maps $T_t(x)$, $t \in [0, 1]$, where each $T_t : \mathcal{X} \to \mathcal{X}$ describes the state of an input at time $t$. We seek the following:

- Initial condition: At $t = 0$, the distribution of transformed points (the push-forward of $\mathbb{P}_-$ by $T_0$) should match or approximate the source distribution $\mathbb{P}_-$.

- Final condition: At $t = 1$, the distribution induced by $T_1$ should match or approximate the target distribution $\mathbb{P}_+$.

Let $v_t(x)$ be a velocity field that describes how each point moves at time $t$. Then, the path-guided CE objective is

$$\min_{(T_t, v_t)} \int_0^1 \int_{\mathcal{X}} c\big(x, v_t(x), t\big)\, dT_t(x)\, dt + \lambda_1 \int_0^1 D_{\chi^2}\big((T_t)_\# \mathbb{P} \,\|\, \mathbb{P}\big) dt$$
$$+ \lambda_2\, D_\psi\big((T_0)_\# \mathbb{P}_- \,\|\, \mathbb{P}_-\big) + \lambda_3\, D_{\chi^2}\big((T_1)_\# \mathbb{P}_- \,\|\, \mathbb{P}_+\big),$$

subject to the continuity equation: $\frac{\partial T_t}{\partial t} + \nabla \cdot (T_t\, v_t) = 0$. Here, $c(x, v_t(x), t)$ is the instantaneous cost of moving point $x$ with velocity $v_t(x)$ at time $t$.

To construct $T_t(x)$, we assume that each point moves along a *constant-speed geodesic* in the feature space $\mathcal{X}$ connecting its original position to its destination under $T_1$. That is, $T_t(x)$ interpolates between $x$ and $T(x)$ such that $T_0(x) = x$ and $T_1(x) = T(x)$. According to results in Villani [2009], such geodesics exist in optimal transport theory, and the time-evolving distribution is then given by $(T_t)_\# \mathbb{P}_-$. For numerical implementation, we use the *back-and-forth algorithm* from Jacobs and Léger [2020], which efficiently approximates dynamic OT paths. Their implementation is publicly available [1]. As a demonstration, we apply path-guided CE to the moons dataset using discrete time steps. The results are shown in Fig. 3, which assumes a high competition cost scenario.

## 4.2 Counterfactual Explanation for Ordered Classifier Families

In conventional CE, eligibility is often determined by a single classifier. In practice, however, such as in loan applications, eligibility may vary with the requested loan amount. For example, a request for $\ell$ might be denied, while a lower amount $\ell'$ could be approved. This necessitates a family of classifiers $h_\ell$, $\ell \in \mathcal{L}$, where each $h_\ell : \mathcal{X} \to \mathcal{Y} = \{\pm 1\}$ makes decisions for loan amount $\ell$. These classifiers follow a natural ordering: $\ell_1 \le \ell_2 \implies h_{\ell_1}(x) \ge h_{\ell_2}(x)$, indicating that eligibility becomes less likely as the loan amount increases.

---

[1] https://github.com/Math-Jacobs/bfm?tab=readme-ov-file)

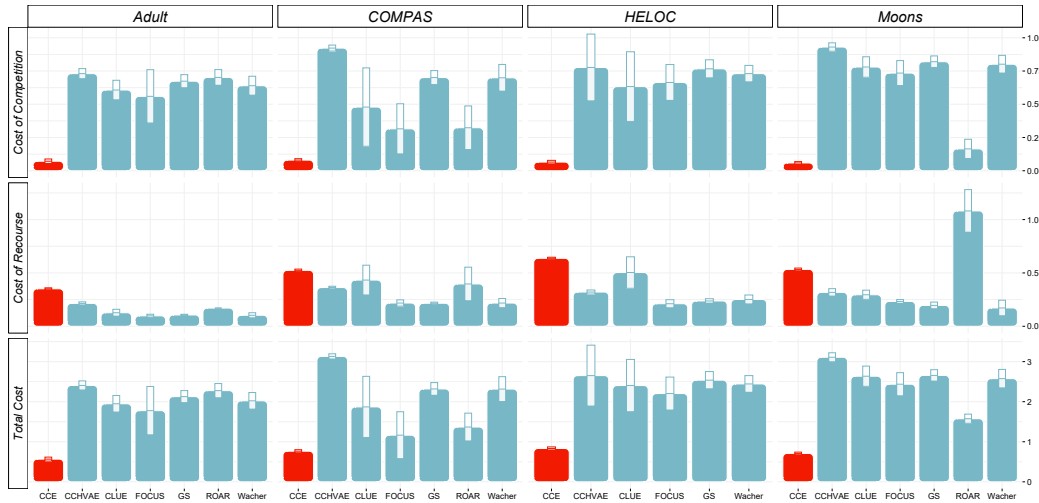

Figure 4: Comparison of modification and competition costs across 100 experiments with different random seeds. Bar plots show average values, while error bars represent standard deviations. Our method (red bar) achieves lower competition cost but not the lowest modification cost, as it moves points toward higher-density regions. However, when considering the combined metric of modification cost and competition efficiency, it outperforms baselines, achieving the best trade-off.

Given this structure, CE seeks a minimal modification $x'$ such that the individual qualifies for a loan of amount at least $\ell$:

$$\mathbf{CE}_\ell(x) = \underset{x' \in \mathcal{X}}{\arg\min} \left\{ c(x, x') \quad \text{s.t.} \quad h_\ell(x') = 1 \right\}.$$

This ensures $x'$ satisfies the eligibility criterion for $\ell$ with minimal cost.

As discussed in Sec. 2, CE can be reframed as an unbalanced OT problem. To extend this to ordered classifiers, we consider the joint distribution $\tilde{\mathbb{P}}$ over features and loan amounts $(x, \ell) \in \mathcal{X} \times \mathcal{L}$. For simplicity, assume $h_\ell(x) = \text{sign}(f(x, \ell))$, where $f : \mathcal{X} \times \mathcal{L} \to \mathbb{R}$ is continuously differentiable and strictly decreasing in $\ell$, i.e., $\frac{\partial f}{\partial \ell} < 0$. The decision boundary is the set $L = \{(x, \ell) \in (\mathcal{X}, \mathcal{L}) : f(x, \ell) = 0\}$. Given that $\nabla f$ is non-zero everywhere, by the implicit function theorem [Lang, 2012, § 5], $L$ forms a $C^1$ $n$-dimensional manifold in the $(n + 1)$-dimensional space $\mathcal{X} \times \mathcal{L}$.

Define $\tilde{\mathbb{P}}_-$ and $\tilde{\mathbb{P}}_+$ as the restrictions of $\tilde{\mathbb{P}}$ to the subsets where $f(x, \ell) < 0$ and $f(x, \ell) > 0$, respectively. The unbalanced CCE framework constructs a recourse map via OT that transports $\tilde{\mathbb{P}}_-$ to $\tilde{\mathbb{P}}_+$ using the cost $c^*((x, \ell), (x', \ell')) = c(x, x')$. To enforce that the resulting loan amount does not decrease, we constrain the OT plan with $1_{\ell' \geq \ell}$. By leveraging tools to solve OT with linear constraints, we can then find efficient recourse maps tailored to ordered classifiers.

## 5 Numerical Studies

In this section, we numerically evaluate our proposed collective CE for algorithmic recourse by comparing it against six baseline approaches: **Wachter** Wachter et al. [2017], **Growing Spheres** Laugel et al. [2017], **CLUE** Antorán et al. [2020], **FOCUS** Lucic et al. [2022], **C-CHVAE** Pawelczyk et al. [2020], and **ROAR** Upadhyay et al. [2021]. We selected a diverse range of algorithms to ensure a broad spectrum of CE methods. We implement baseline methods using the open-source CARLA Pawelczyk et al. [2021] (Counterfactual And Recourse Library) framework in Python, which offers standardized interfaces for generating counterfactual explanations and recourse interventions. Our experimental code will be released after review. We conducted experiments on three real-world datasets commonly used in the literature to evaluate recourse methods: **Adult**Becker and Kohavi [1996], **COMPAS**Angwin et al. [2016], and **HELOC**FICO [2018], as well as one synthetic dataset, **Moons**Pedregosa et al. [2011], which is also frequently utilized in illustrating algorithmic recourse.

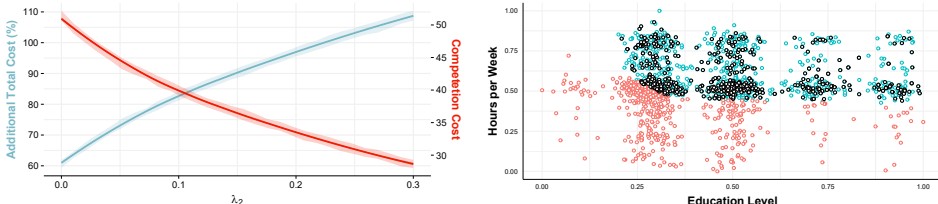

Figure 5: (Left) The blue curve represents the percentage increase in modification cost of CCE relative to standard CE as $\lambda_2$ varies from 0.01 to 0.3. The red curve illustrates the competition cost obtained by $\lambda_2 D_{\chi^2}(T_\# \mathbb{P}_- \parallel \mathbb{P}_+)$. Both curves are supported with confidence intervals. As expected, there is a trade-off between modification and competition cost measures. (Right) The result of CCE on the Adult dataset with $\lambda_2 = 0.1$.

We specifically employed the Moons dataset because it is two-dimensional, allowing us to explicitly demonstrate the features and highlight the differences between our method and other baselines.

The real data are derived from the CARLA package and preprocessed for each dataset (for more details, see Appendix F). We use only two continuous actionable features to determine the best recourse. Each dataset is randomly split into training (80%) and test (20%) sets. Two non-linear methods, Multilayer Perceptron or Random Forest (corresponding method), are trained on each dataset to serve as predictive models for which we seek algorithmic recourse. Baseline hyperparameters are tuned according to the guidelines in the CARLA documentation or prior literature (see Table 1). To compute the CCE, we employed the algorithm in Sec. 2.4. We also use the $\ell_2$ Euclidean as a cost.

To construct $\mathbb{P}_-$ and $\mathbb{P}_+$, we select or generate 1000 instances for each label, negative or positive, and construct the sample sets $\mathcal{D}^+$ and $\mathcal{D}^-$. In each experiment, we run baseline methods and compute the average modification cost, $\frac{1}{n}\sum_{i=1}^n c(x_i, \mathbf{CE}(x_i))$. Additionally, we compute $\lambda_2 D_{\chi^2}(T_\#\mathbb{P}_- \parallel \mathbb{P}_+)$ to measure the divergence between the transported distribution and the target distribution. In our experiment, we put $\lambda_2 = 0.1$. To evaluate the competition cost, we discretize the space into a grid and determine the proportions $p_i$ and $q_i$ of samples corresponding to $\mathcal{D}^+$ and $\mathbf{CE}(\mathcal{D}^-)$ within each grid cell. Finally, the competition cost is computed as $\lambda_2 \sum_{i=1}^n \frac{(q_i - p_i)^2}{p_i}$. This approach allows for a structured comparison of recourse effectiveness and cost across different methods.

We conducted 100 experiments, each with a different random seed, and computed both the modification and competition cost metrics for each run. In Fig. 4, the average results across all experiments are represented by bar plots, while the standard deviation is illustrated using error bars. As expected, our method demonstrates lower competition cost compared to others. However, its modification cost is not the lowest, as it tends to move points toward higher-density regions. Nevertheless, when considering the combined metric of modification cost plus competition efficiency ($\frac{1}{n}\sum_{i=1}^n c(x_i, \mathbf{CE}(x_i)) + \lambda_2 \sum_{i=1}^n \frac{(q_i - p_i)^2}{p_i}$), our method outperforms the baselines, offering the most balanced trade-off.

Another simulation explores the role of $\lambda_2$, the competition cost. We investigate the behavior of the CCE modification and competition as $\lambda_2$ varies within the range $[0.01, 0.3]$. As expected, and as illustrated by the simulation in Fig. 5 left, there is a trade-off between modification and competition costs. By tuning $\lambda_2$ in real applications, we can adjust the relative weight of modification and competition costs to achieve a more realistic recourse.

To find the temporal path for each recourse, as explained in Sec. 4.1, we employed the back-and-forth method (see Appendix E.3) to determine the optimal temporal curve from the initial state to the modified resource. We demonstrate the construction of path-guided CE for the moons dataset based on discrete temporal steps. The results are illustrated in Fig. 3. In the right figure, the optimal flow that transfers the negative samples into the positive area is depicted over four steps.

Finally, to observe the CCE recourse, after determining the optimal plan, we randomly select the best state using a multinomial distribution based on the probabilities derived from the optimal plan matrix. The results are presented for the Moons dataset in Fig. 1, the Adult dataset in the right panel of Fig. 5, and the COMPAS and HELOC datasets in Fig. 6.

# 6 Further Related Work

In Poyiadzi et al. [2020], Kanamori et al. [2020], population probability is encoded into the cost function, guiding CE toward denser regions near the boundary $L$. While this helps mitigate outlier recommendations, these methods remain individual-centric and prone to overcrowding. Moreover, their integration of density lacks a principled foundation, appearing somewhat arbitrary.

Group counterfactual explanations improve interpretability and fairness by addressing multiple instances simultaneously. Warren et al. [2023] developed an algorithm for high-coverage, model-faithful explanations, enhancing user understanding. Wielopolski et al. [2024] introduced a gradient-based method linking local, group, and global counterfactuals. Carrizosa et al. [2024b] proposed optimization models minimizing perturbation costs with linking constraints. Lodi and Ramírez-Ayerbe [2024] presented a column generation framework for sparse, scalable group explanations. Fragkathoulas et al. [2024] developed a graph-based approach ensuring feasible, fair group counterfactuals via subgroup formation.

In Tsirtsis and Gomez Rodriguez [2020], utility functions represent decision-makers' objectives, guiding the optimization of policies and CE to maximize desired outcomes in strategic settings. Recently, Carrizosa et al. [2024a] introduced a notion of collective CE, focusing on optimizing the modification cost for a group of instances rather than for individuals. This approach aims to harmonize the behavior of CE within a group, thereby mitigating the occurrence of cost-outlier CE. However, by overlooking the underlying density, this method might still result in CEs that are outliers concerning probability measures. Additionally, coupling CE within a group could amplify the externalities.

To the best of our knowledge, our method is novel in the literature, employing population dynamics to transform the conventional CE problem into a collective version. This approach is more realistic and addresses certain issues inherent in conventional CE.

Our work is also indirectly related to strategic classification Hardt et al. [2016]. Both CE and strategic classification involve individuals seeking a positive label. CE recommends cost-minimizing actions under a fixed classifier, which is incentive-compatible (IC). In contrast, strategic classification accounts for individuals strategically altering features, with the classifier adapting accordingly. Our work adds a new layer of realism to CE: individuals taking similar actions may face competition. This effect has recently been modeled as an externality in strategic classification Hossain et al. [2024]. Competition challenges the standard IC assumption in CE, as cost-minimizing recommendations may no longer ensure incentive compatibility. While our framework does not explicitly model deviations from recommendations, it focuses on maximizing social welfare in the presence of externalities. Extending it to account for strategic deviations would be an interesting direction for future work.

# 7 Discussion and Future works

CCE is designed to address the limitations of standard CE by incorporating the population distribution. By accounting for societal competition for favorable outcomes, CCE mitigates performative effects arising from large-scale behavioral shifts that affect the cost function. Both theory and numerical studies demonstrate CCE's effectiveness in tackling core challenges in counterfactual design.

Importantly, computing the CCE solution does not require estimating $\mathbb{P}$; our methods work directly with samples, offering flexibility. No assumption is needed on $h$, allowing compatibility with black-box classifiers.

Causality is central to algorithmic recourse, as interventions on some features can causally affect others Karimi et al. [2020]. Our framework supports causal recourse through the use of causally fair metrics Ehyaei et al. [2024a,b, 2023a], which map from endogenous to exogenous variables, thereby removing direct dependencies before applying CCE. We omit technical details here to maintain focus on collective CE, as full integration into a causal framework warrants separate study.

Choosing $\lambda_2$ is application-specific and non-trivial. While this work does not address general models of competition in recourse, we focus on competition over externalities Altmeyer et al. [2023] using a concrete model. Future work should explore broader notions of externalities in this context.

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

# A    Supplementary Materials

## A.1    Definitions

**Definition 1 (Push-forward Measure)** *Let $\mathbb{P}$, $\mathbb{Q}$ be two probability measures in $\mathcal{P}(\mathbb{R}^d)$ and $T :$ $\mathbb{R}^d \to \mathbb{R}^d$ is map, the measure $\mathbb{Q}$ is called the push-forward of $\mathbb{P}$ through $T$ is denoted by $T_\# P$ if:*

$$\mathbb{Q}(B) = \mathbb{P}(T^{-1}(B)), \quad \forall B \subset \mathbb{R}^d$$

**Definition 2 (Weak Topology)** *Weak topology on a space of probability measures on $\mathbb{R}^d$, denoted by $\mathcal{P}(\mathbb{R}^d)$, is defined by convergence in distribution. A sequence of probability measures $(\mathbb{P}_n)_{n \in \mathbb{N}}$ in $\mathcal{P}(\mathbb{R}^d)$ converges weakly to a probability measure $\mathbb{P}$ if for every bounded continuous function $f : \mathbb{R}^d \to \mathbb{R}$,*

$$\lim_{n \to \infty} \mathbb{E}_{\mathbf{X}_n \sim \mathbb{P}_n} [f(\mathbf{X}_n)] = \mathbb{E}_{\mathbf{X} \sim \mathbb{P}}[f(\mathbf{X})].$$

**Definition 3 (Set of Couplings)** *The set $\Gamma(\mathbb{P}, \mathbb{Q})$ represents the couplings of probability distributions $\mathbb{P}, \mathbb{Q} \in \mathcal{P}(\mathbb{R}^d)$, comprising distributions over $\mathbb{R}^d \times \mathbb{R}^d$ with margins $\mathbb{P}$ and $\mathbb{Q}$. A measure $\pi$ belongs to $\Gamma(\mathbb{P}, \mathbb{Q})$ if and only if*

$$\pi(A \times \mathbb{R}^d) = \mathbb{P}(A) \quad and \quad \pi(\mathbb{R}^d \times B) = \mathbb{Q}(B) \quad \forall A, B \subset \mathbb{R}^d$$

*By extension, a random pair $(X, Y) \sim \pi$, where $\pi \in \Gamma(\mathbb{P}, \mathbb{Q})$, will also be called a coupling of $\mathbb{P}$ and $\mathbb{Q}$.*

**Definition 4 (Linear Constraints)** *Let $\mathcal{C}_L(\mathbb{P})$ and $\mathcal{C}_L(\mathbb{Q})$ be the space of continuous functions on $\mathcal{X}$ and $\mathcal{X}'$ and $L^1$-integrable respect to $\mathbb{P}$ and $\mathbb{Q}$ measures. Let $\mathcal{C}_L(\mathbb{P}, \mathbb{Q})$ be the family of continuous functions on $\mathcal{X} \times \mathcal{X}'$ such that:*

$$\mathcal{C}_L(\mathbb{P}, \mathbb{Q}) = \{h \in \mathcal{C}(\mathcal{X} \times \mathcal{X}') : \exists f = f_1 + f_2 \ s.t. \ |h| \leq f\},$$

*where $f_1 \in \mathcal{C}_L(\mathbb{P})$, $f_2 \in \mathcal{C}_L(\mathbb{Q})$.*

**Definition 5 (The Wasserstein Metric)** *In the metric space $(\mathcal{X}, d)$, OT naturally defines a metric known as the Wasserstein or Earth Mover's distance. To define the Wasserstein metric, we consider the space of probability measures with finite $p$-th moment (Wasserstein space):*

$$\mathcal{P}_q(\mathcal{X}) = \{\mathbb{P} \in \mathcal{P}(\mathcal{X}) : \mathbb{E}_{\mathbf{X} \sim \mathbb{P}}[d^q(\mathbf{X}, x')] < \infty, \forall x' \in \mathcal{X}\}$$

*For two probability measures $\mathbb{P}, \mathbb{Q} \in \mathcal{P}_q(\mathcal{X})$, the $\mathbb{P}$-Wasserstein distance is defined as:*

$$W_q(\mathbb{P}, \mathbb{Q}) = \left( \inf_{\pi \in \Gamma(\mathbb{P}, \mathbb{Q})} \mathbb{E}_{(\mathbf{X}, \mathbf{X}') \sim \pi}[d^q(\mathbf{X}, \mathbf{X}')] \right)^{\frac{1}{q}}.$$

*This metric is positive-definite, finite, symmetric, and adheres to the triangle inequality [Villani, 2009, § 6].*

**Definition 6 (Truncated Probability Measure)** *Let $\mathbb{P} \in \mathcal{P}(\mathbb{R}^d)$ be a probability measure and $A \subseteq \mathbb{R}^d$ be Borel subset of $\mathbb{R}^d$ such that $\mathbb{P}(A) > 0$, the truncated probability measure $\mathbb{P}_A \in \mathcal{P}(\mathbb{R}^d)$ is defined as:*

$$\mathbb{P}_A(B) = \frac{\mathbb{P}(A \cap B)}{\mathbb{P}(A)}$$

*for all $B \in \mathcal{B}(\mathbb{R}^d)$.*

**Definition 7 (lower semi-continuous)** *A function $f : \mathbb{R}^d \to \mathbb{R} \cup \{+\infty\}$ is said to be lower semi-continuous (l.s.c) at a point $x_0 \in rd$ if for every $\epsilon > 0$ there exists a neighborhood $U$ of $x_0$ such that for all $x \in U$, $f(x) > f(x_0) - \epsilon$. In economic terms, this concept implies that small perturbations in the input do not lead to a substantial decrease in the function value, signifying a form of stability or predictability in economic models, such as cost functions in production processes.*

**Definition 8 (Closure)** *Given a set $A$ in a topological space $X$, the* closure *of $A$, denoted by $\overline{A}$, is the smallest closed set in $X$ that contains $A$. Equivalently, it includes all the points of $A$ along with all its limit points (i.e., points that can be approached arbitrarily closely by points in $A$).*

**Definition 9 (Continuous Measure)** *A measure $\mu$ on a measurable space $(X, \mathcal{F})$ is said to be continuous if for every $A \in \mathcal{F}$, $\mu(A) = 0$ whenever $A$ is a set of a single point. In other words, $\mu(\{x\}) = 0$ for every $x \in X$.*

**Definition 10 ($\varphi$-divergence)** *The $\varphi$-divergence between two probability distributions $\mathbb{P}$ and $\mathbb{Q}$ over the same probability space, for a convex function $\varphi$, is defined as*

$$D_\varphi(\mathbb{P}||\mathbb{Q}) = \int \varphi\left(\frac{d\mathbb{P}}{d\mathbb{Q}}\right) d\mathbb{Q}$$

*where $\frac{d\mathbb{P}}{d\mathbb{Q}}$ is the Radon-Nikodym derivative of $\mathbb{P}$ with respect to $\mathbb{Q}$. The divergence measures the difference between the two distributions, with different choices of $\varphi$ leading to different divergence measures. Common examples include the Kullback-Leibler divergence for $\varphi(x) = x \log x$, the Total Variation distance for $\varphi(x) = \frac{1}{2}|x - 1|$, and the squared Hellinger distance for $\varphi(x) = (\sqrt{x} - 1)^2$.*

**Definition 11 (n-dimensional Hausdorff Measure)** *Let $(X, d)$ be a metric space. The n-dimensional Hausdorff measure $\mathcal{H}^n$ of a subset $A \subseteq X$ is defined as follows:*

$$\mathcal{H}^n(A) = \lim_{\delta \to 0} \inf\left\{\sum_{i=1}^{\infty} (\mathrm{diam}(U_i))^n : \{U_i\} \text{ is a } \delta\text{-cover of } A\right\}$$

*where a $\delta$-cover of $A$ is a countable collection of sets $\{U_i\}$ with $\mathrm{diam}(U_i) < \delta$ such that $A \subseteq \bigcup_i U_i$, and $\mathrm{diam}(U_i)$ is the diameter of the set $U_i$. For $n \in \mathbb{N}$, $\mathcal{H}^n$ generalizes the notion of n-dimensional volume, with $\mathcal{H}^1$ representing length, $\mathcal{H}^2$ area, and $\mathcal{H}^3$ volume.*

**Definition 12 ($\delta$-Confidence Positive Region)** *$\delta$-confidence positive region is denoted by $L_\delta^+$ consists of smallest buffer of $L$ in $\mathcal{X}^+$ i.e., $B = \{x \in \mathcal{X}^+ : dist(x, L) \leq r\}$ such that $\mathbb{P}(B) \geq \epsilon$. where $dist(x, L) := \inf\{c(x, x') : x' \in L\}$.*

### A.2   Lemmas and Theorems

**Lemma 1 (Measurability of the Recourse Map)** *Suppose $\mathcal{X}$ is a standard Borel space, $h : \mathcal{X} \to \mathcal{Y}$ is a measurable function, and $c : \mathcal{X} \times \mathcal{X} \to \mathbb{R}$ is a jointly measurable cost. Define*

$$R(x) := \operatorname*{arg\,min}_{x' \in \mathcal{X}^+} \{c(x, x')\}.$$

*Then, $R$ is a measurable set-valued map. If $R(x)$ is non-empty and closed for all $x \in \mathcal{X}^-$, there exists a measurable $T : \mathcal{X}^- \to \mathcal{X}^+$ with $T(x) \in R(x)$ for all $x$.*

## B   Overview of Optimal Transport and Extensions

This section provides an overview of OT and its extensions utilized in this study. OT, initially introduced by Monge [1781], focused on cost-efficient transportation of soil for fortifications. Generally, OT aims to transfer probability measures from a space $\mathcal{X}$ to $\mathcal{X}'$. While $\mathcal{X}$ and $\mathcal{X}'$ are typically Polish spaces, in this work, they are considered open or closed bounded subsets of $\mathbb{R}^d$.

### B.1   Monge Problem

Let $\mathbb{P} \in \mathcal{P}(\mathcal{X}), \mathbb{Q} \in \mathcal{P}(\mathcal{X}')$ be two probability measures, and let $c : \mathcal{X} \times \mathcal{X}' \to \mathbb{R}$ be a ground-cost function representing the cost of transporting a unit mass from $x$ to $y$. The *Monge's OT problem* is to find a map $T : \mathcal{X} \to \mathcal{X}'$ that pushes forward $\mathbb{P}$ to $\mathbb{Q}$ (i.e., $T_\#\mathbb{P} = \mathbb{Q}$) and minimizes transportation effort:

$$\min_{T: T_\#\mathbb{P}=\mathbb{Q}} \quad \mathbb{E}_{\mathbf{X} \sim P}[c(\mathbf{X}, T(\mathbf{X}))]. \tag{7}$$

An *optimal map* is a $T$ that minimizes this objective. A function $T$ satisfying $T_\#\mathbb{P} = \mathbb{Q}$ is called a *push-forward map* [Ambrosio et al., 2021, § 1.2].

Regardless of the cost function $c$, Monge's problem may be ill-posed due to the nonexistence of a push-forward map and weak sequential closure issues w.r.t. the weak topology Ambrosio et al. [2013]. After 150 years, Kantorovich [1942] addressed these limitations by relaxing the problem.

## B.2 Kantorovich Problem

Rather than finding an optimal map, Kantorovich proposed minimizing transportation cost for coupling $\pi \in \Gamma(\mathbb{P}, \mathbb{Q})$:

$$\min_{\pi \in \Gamma(\mathbb{P}, \mathbb{Q})} \quad \mathbb{E}_{(\mathbf{X}, \mathbf{X}') \sim \pi}[c(\mathbf{X}, \mathbf{X}')]. \tag{8}$$

The solution of the Kantorovich problem, when it exists, is called the *optimal plan* [Ambrosio et al., 2021, § 2.1]. The set of push-forward maps, denoted $\Gamma_0(\mathbb{P}, \mathbb{Q})$, known as Monge couplings, are special cases of couplings characterized by $\pi \sim (X, T(X))$.

## B.3 Kantorovich–Rubinstein Duality

Kantorovich reformulated OT as a convex problem on $\mathcal{P}(\mathcal{X} \times \mathcal{X}')$, with its dual expressed as a constrained concave maximization problem [Ambrosio et al., 2021, § 3.1]. Kantorovich duality states that the minimum of the Kantorovich problem equals the maximum of the dual problem over bounded and continuous *Kantorovich potentials* $\varphi : \mathcal{X} \to \mathbb{R}$ and $\psi : \mathcal{X}' \to \mathbb{R}$:

$$\sup_{(\varphi, \psi) \in \Phi_c} \quad \mathbb{E}_{\mathbf{X} \sim P}[\varphi(\mathbf{X})] + \mathbb{E}_{\mathbf{X}' \sim Q}[\psi(\mathbf{X}')], \tag{9}$$

subject to $\Phi_c = \{(\varphi, \psi) : \varphi(x) + \psi(x') \leq c(x, x')\}$.

For example, consider a logistics company transporting goods from $x$ to $x'$. The company sets a loading fee $\varphi(x)$ at $x$ and an unloading fee $\psi(x')$ at $x'$. Their profit margin $\varphi(x) - \psi(x')$ must not exceed the transportation cost $c(x, x')$. To maximize profits, the company adjusts the pricing functions $\varphi$ and $\psi$ in Eq. (9) (see Galichon, 2018, § 2 for more examples).

## B.4 Optimal Transport with Linear Constraints

In practical applications, solutions to the OT problem often need to satisfy constraints. Zaev [2015] incorporated these by adding linear constraints. The constrained OT problem seeks optimal couplings with additional conditions over the family $\mathcal{W} \subset \mathcal{C}_L(\mathbb{P}, \mathbb{Q})$ of continuous and $L^1$-integrable functions on $\mathcal{X} \times \mathcal{X}'$:

$$\min_{\pi \in \Gamma(\mathbb{P}, \mathbb{Q})} \quad \mathbb{E}_{(\mathbf{X}, \mathbf{X}') \sim \pi}[c(\mathbf{X}, \mathbf{X}')] \quad \text{s.t.} \quad \mathbb{E}_{(\mathbf{X}, \mathbf{X}') \sim \pi}[w(\mathbf{X}, \mathbf{X}')] = 0, \quad \forall w \in \mathcal{W}. \tag{10}$$

Invariant and martingale OT are examples of such constrained problems.

## B.5 Unbalanced Optimal Transport (UOT)

Conventional OT assumes total supply equals total demand. UOT extends OT to scenarios where source and target distributions differ in total mass, incorporating terms for creation and annihilation of mass Séjourné et al. [2022]. Let $\mu \in \mathcal{M}_+(\mathcal{X}), \nu \in \mathcal{M}_+(\mathcal{X}')$ be two positive measures and $\pi \in \mathcal{M}_+(\mathcal{X} \times \mathcal{X}')$. The UOT problem is:

$$\min_{\pi \in \mathcal{M}_+(\mathcal{X}, \mathcal{X}')} \int_{\mathcal{X} \times \mathcal{X}'} c(x, x') \, d\pi(x, x') + \lambda_1 D_{\varphi_1}(\mu, \pi_1) + \lambda_2 D_{\varphi_2}(\nu, \pi_2),$$

where $D_{\varphi_1}$ and $D_{\varphi_2}$ are $\varphi-$divergence terms for mass creation and annihilation, with $(\lambda_1, \lambda_2)$ as hyper-parameters and $(\pi_1, \pi_2)$ as marginals of $\pi$.

## B.6 Dynamic Optimal Transport (DOT)

DOT extends OT by incorporating a temporal dimension to model mass evolution Benamou and Brenier [2000]. Let $\mu_t(\mathbf{X}) = \mu(t, \mathbf{X})$ represent a path of probability measures such that $\mu_0 = \mathbb{P}$ and $\mu_1 = \mathbb{Q}$, and $v_t(x) = v(t, \mathbf{X})$ be a velocity field. DOT is formulated as:

$$\arg\min_{(\mu_t, v_t)} \left\{ \int_0^T \int_{\mathbb{R}^n} c(x, v_t(x, t), t) \, d\mu_t(x) \, dt \right\},$$

subject to the continuity equation $\partial \mu_t / \partial t + \nabla \cdot (\mu_t v_t) = 0$. Here, $c(x, x', t)$ represents the transport cost at time $t$, and the integral computes the total transport cost.

### B.7 Fundamental Theorem of Optimal Transport

**Theorem 1 (Fundamental Theorem of Optimal Transport)** *Assume $c : \mathcal{X} \times \mathcal{X}' \to \mathbb{R}$ is continuous, bounded below, and let $\mu \in \mathcal{P}(\mathcal{X}), \nu \in \mathcal{P}(\mathcal{X}')$ satisfy $c(x, x') \leq a(x) + b(x')$, for $a \in L^1(\mu), b \in L^1(\nu)$. For $\pi \in \Gamma(\mu, \nu)$, the following are equivalent:*

- *$\pi$ is optimal,*

- *The minimum of the Kantorovich problem equals the supremum of the dual problem* (9)*, attained by $(\varphi, \psi)$ of the form $(\varphi, \varphi^{c+})$ for some c-concave function $\varphi$.*

## C  A Mean-Field Population Model: Explanation and Solutions

We consider the PDE

$$\frac{\partial \mathbb{U}}{\partial t}(x, t) \ = \ \nabla \cdot \Big( \mathbb{U}(x, t) \nabla \big[ \beta \, \mathbb{U}(x, t) \ - \ \alpha \, \mathbb{S}(x) \big] \Big), \tag{11}$$

where $\mathbb{U}(x, t) \geq 0$ is the population density, $\mathbb{S}(x)$ is a static resource function, and $\alpha, \beta > 0$. Equation (11) arises from a continuity equation with velocity

$$v \ = \ -\nabla[\beta \, \mathbb{U} - \alpha \, \mathbb{S}] \ = \ \alpha \, \nabla \mathbb{S} \ - \ \beta \, \nabla \mathbb{U},$$

indicating attraction toward regions of higher $\mathbb{S}(x)$ and repulsion from high-density regions $\mathbb{U}(x, t)$. Equivalently, one may view $\beta \, \mathbb{U} - \alpha \, \mathbb{S}$ as a local potential; a gradient-flow formulation yields the same PDE.

A steady-state $\mathbb{U}^*(x)$ satisfies

$$\nabla \cdot \Big( \mathbb{U}^*(x) \nabla [\beta \, \mathbb{U}^*(x) - \alpha \, \mathbb{S}(x)] \Big) \ = \ 0.$$

A simple family of solutions has

$$\beta \, \mathbb{U}^*(x) - \alpha \, \mathbb{S}(x) \ = \ C \quad \Longrightarrow \quad \mathbb{U}^*(x) \ = \ \frac{\alpha}{\beta} \, \mathbb{S}(x) \ + \ \frac{C}{\beta}.$$

Boundary conditions or normalization (e.g., total mass) determine $C$. Since we suppose $\mathbb{U}^*$ is the density of the distribution, we have $C = 0$.

## D  Proofs

### D.1  Proof of Lemma 1

We proved the proposition in a more general case. Define the set-valued map $A(x) := \{x' \in \mathcal{X} : h(x') \neq h(x)\}$. Since $h$ is measurable, the set $\{(x, x') : h(x') = h(x)\}$ is a measurable subset of $\mathcal{X} \times \mathcal{X}$. Consequently, its complement $\{(x, x') : h(x') \neq h(x)\} = \{(x, x') : x' \in A(x)\}$ is also measurable. Hence, $A(\cdot)$ is a measurable set-valued map. Next, define the extended cost function.

$$g(x, x') := \begin{cases} c(x, x') & \text{if } x' \in A(x), \\ +\infty & \text{otherwise.} \end{cases}$$

Since $c$ is jointly measurable and $A(\cdot)$ is measurable, the function $g$ is measurable on $\mathcal{X} \times \mathcal{X}$. Thus, for each fixed $x$, the minimization problem $R(x) = \arg\min_{x'} g(x, x')$ is well-defined.

We assume conditions ensuring the existence of a minimizer. For instance, if $X$ is compact (in a suitable topological setting) and $c$ is lower semicontinuous and coercive, then for each $x \in X$, the set $R(x)$ is non-empty and closed. These standard conditions are often satisfied in algorithmic recourse scenarios, where one restricts attention to compact feasible domains or ensures appropriate behavior of $c$.

Now, $R(x)$ is obtained as the set of minimizers of a measurable function $g(x, x')$ over a measurable and closed-valued set $A(x)$. Standard results from measurable selection theory (e.g., the Kuratowski–

Ryll-Nardzewski measurable selection theorem Bogachev and Ruas [2007]) ensure that a measurable selection from $R(x)$ exists provided $R(x)$ is non-empty and closed.

Specifically, the Kuratowski–Ryll-Nardzewski theorem states that if $\mathcal{X}$ is a standard Borel space and $T : \mathcal{X} \rightrightarrows \mathcal{X}$ is a measurable set-valued map with non-empty closed (or compact) values, then there exists a measurable function $T : \mathcal{X} \to \mathcal{X}$ such that $T(x) \in R(x)$ for all $x$. Applying this theorem to our setting, we obtain such a measurable selection $T$. Therefore, under the stated conditions, $R(\cdot)$ is measurable in the sense that it admits a measurable selection, completing the proof.

### D.2 Proof of Prop. 1

To prove it is sufficient to show that:
$$D_{\chi^2}\left(\mathbb{P}_{\mathbf{CE}} \,\|\, \mathbb{P}_+\right) = \lambda^2 \, D_{\chi^2}\left(T_\#\mathbb{P}_- \,\|\, \mathbb{P}_+\right) \tag{12}$$
Recall that the $\chi^2$-divergence of a measure $T_\#\mathbb{P}$ w.r.t. $\mathbb{P}_+$ is defined as
$$D_{\chi^2}(T_\#\mathbb{P}_- \,\|\, \mathbb{P}_+) = \int \left(\frac{dT_\#\mathbb{P}_-}{d\mathbb{P}_+}(y) - 1\right)^2 d\mathbb{P}_+(y),$$
provided $T_\#\mathbb{P}_- \ll \mathbb{P}_+$. Now consider the mixture $\lambda T_\#\mathbb{P}_- + (1-\lambda)\mathbb{P}_+$. Its density w.r.t. $\mathbb{P}_+$ is
$$\frac{d(\lambda T_\#\mathbb{P}_- + (1-\lambda)\mathbb{P}_+)}{d\mathbb{P}_+}(y) = \lambda\frac{dT_\#\mathbb{P}_-}{d\mathbb{P}_+}(y) + (1-\lambda)\cdot 1 = 1 + \lambda\left(\frac{dT_\#\mathbb{P}_-}{d\mathbb{P}_+}(y) - 1\right).$$
Substituting this expression into the definition of $\chi^2$-divergence, we get
$$\begin{aligned}
D_{\chi^2}\left(\lambda T_\#\mathbb{P}_- + (1-\lambda)\mathbb{P}_+ \,\|\, \mathbb{P}_+\right) &= \int \left(\left[1 + \lambda\left(\frac{dT_\#\mathbb{P}_-}{d\mathbb{P}_+}(y) - 1\right)\right] - 1\right)^2 d\mathbb{P}_+(y) \\
&= \int \left(\lambda\left(\frac{dT_\#\mathbb{P}_-}{d\mathbb{P}_+}(y) - 1\right)\right)^2 d\mathbb{P}_+(y) \\
&= \lambda^2 \int \left(\frac{dT_\#\mathbb{P}_-}{d\mathbb{P}_+}(y) - 1\right)^2 d\mathbb{P}_+(y),
\end{aligned}$$
as required.

### D.3 Proof of Prop. 2

The optimization is over $\pi \in \mathcal{P}(\mathcal{X} \times \mathcal{X})$ such that its first marginal equals $\mathbb{P}_-$. This implies $\pi_1 = \mathbb{P}_-$. Since $\mathbb{P}_-$ is fixed, any admissible $\pi$ must have this prescribed marginal. Consider a minimizing sequence $(\pi_n)_{n \in \mathbb{N}}$ such that
$$\left(\int c^q d\pi_n\right)^{1/q} + \eta\lambda^2 D_{\chi^2}(\pi_{n,2}\|\mathbb{P}_+) \to \inf_{\pi:\pi_1=\mathbb{P}_-}\left\{\left(\int c^q d\pi\right)^{1/q} + \eta\lambda^2 D_{\chi^2}(\pi_2\|\mathbb{P}_+)\right\}.$$

If $(\pi_n)$ attempted to "push mass to infinity," the cost term $\int c^q d\pi_n$ would either blow up or, if bounded, the $\chi^2$-divergence term would penalize deviations of $\pi_{n,2}$ from $\mathbb{P}_+$ significantly. In other words, the $\chi^2$-penalty encourages $\pi_{n,2}$ to remain close to $\mathbb{P}_+$, and the cost term controls the large-scale displacement. Together, these terms prevent the mass from escaping, ensuring that $(\pi_n)$ is tight. By Prokhorov's theorem Billingsley [2013], there exists a subsequence that converges weakly to some $\pi \in \mathcal{P}(\mathcal{X} \times \mathcal{X})$, it means $\pi_n \rightharpoonup \pi$. Weak convergence and the linearity of projection imply that the marginals also converge weakly. Since each $\pi_n$ satisfies $\pi_{n,1} = \mathbb{P}_-$, and $\mathbb{P}_-$ is fixed, the continuity of the marginalization map ensures $\pi_1 = \mathbb{P}_-$.

The objective function is:
$$J(\pi) = \left(\int_{\mathcal{X} \times \mathcal{X}} c(x,y)^q \, d\pi(x,y)\right)^{1/q} + \eta\lambda^2 D_{\chi^2}(\pi_2\|\mathbb{P}_+).$$

The map $\pi \mapsto \int c^q d\pi$ is linear in $\pi$, and $c^q$ is lower semicontinuous. By the Portmanteau theorem Billingsley [2013], for $\pi_n \rightharpoonup \pi$:
$$\int c^q d\pi \leq \liminf_{n\to\infty} \int c^q d\pi_n.$$

Since $z \mapsto z^{1/q}$ is continuous and increasing, we have:

$$\left( \int c^q d\pi \right)^{1/q} \leq \liminf_{n \to \infty} \left( \int c^q d\pi_n \right)^{1/q}.$$

For the $\chi^2$-divergence, $D_{\chi^2}(\cdot \| \mathbb{P}_+)$ is lower semicontinuous with respect to weak convergence of measures. Thus, as $\pi_{n,2} \rightharpoonup \pi_2$:

$$D_{\chi^2}(\pi_2 \| \mathbb{P}_+) \leq \liminf_{n \to \infty} D_{\chi^2}(\pi_{n,2} \| \mathbb{P}_+).$$

Combining these, we have:

$$J(\pi) \leq \liminf_{n \to \infty} J(\pi_n).$$

Thus, $J(\cdot)$ is lower semicontinuous w.r.t. weak convergence. Since $(\pi_n)$ is a minimizing sequence, we have by definition:

$$\liminf_{n \to \infty} J(\pi_n) = \inf_{\pi : \pi_1 = \mathbb{P}_-} J(\pi).$$

By the lower semicontinuity established above:

$$J(\pi) \leq \liminf_{n \to \infty} J(\pi_n) = \inf_{\pi' : \pi'_1 = \mathbb{P}_-} J(\pi').$$

Hence, $\pi$ attains the infimum:

$$J(\pi) = \inf_{\pi : \pi_1 = \mathbb{P}_-} J(\pi).$$

This shows that a solution $\pi^*$ exists.

### D.4 Proof of Prop. 3

Let $\pi_*$ be the optimal solution of (5), and let $\pi_{\lambda_1}$ be the solution to (6). Since $\pi_*$ has $\pi_{*,1} = \mathbb{P}_-$, plugging $\pi_*$ into the relaxed problem's objective shows

$$(\mathbb{E}_{(x,y) \sim \pi_*}[c(x,y)^q])^{1/q} + \lambda_2 \, D_{\chi^2}(\pi_{*,2} \| \mathbb{P}_+) \;\geq\; (\mathbb{E}_{(x,y) \sim \pi_{\lambda_1}}[c(x,y)^q])^{1/q} + \lambda_1 \, D_\psi(\pi_{\lambda_1,1} \| \mathbb{P}_-) + \lambda_2 \, D_{\chi^2}(\pi_{\lambda_1,2} \| \mathbb{P}_+).$$

As $\lambda_1 \to \infty$, any deviation of $\pi_{\lambda_1,1}$ from $\mathbb{P}_-$ would make the divergence term $\lambda_1 D_\psi(\pi_{\lambda_1,1} \| \mathbb{P}_-)$ unbounded. Hence, $\pi_{\lambda_1,1} \to \mathbb{P}_-$.

By tightness and lower semicontinuity arguments, any limit point of $\{\pi_{\lambda_1}\}$ has first marginal $\mathbb{P}_-$ and cannot exceed the minimal value of (5). Thus, $\pi_{\lambda_1} \to \pi_*$, completing the proof.

### D.5 Proof of Prop. 4

To establish the time complexity of the Projected-Gradient CCE Solver in Algorithm 1, observe that each iteration from lines 3–9 involves the following steps:

- **Marginal computations** (line 3). Computing $\pi_1^{(t)}(i) = \sum_{j=1}^n \pi^{(t)}(i,j)$ for all $i = 1, \ldots, m$ takes $O(mn)$ operations. Similarly, computing $\pi_2^{(t)}(j) = \sum_{i=1}^m \pi^{(t)}(i,j)$ for all $j = 1, \ldots, n$ also takes $O(mn)$ operations. Overall, marginal updates require $O(mn)$ time.

- **Gradient computation** (line 4). We compute $\nabla_{\pi(i,j)} F$ for each pair $(i,j)$, involving only a constant number of arithmetic and logarithmic operations. Hence, the gradient calculation for all $m \times n$ entries is $O(mn)$.

- **Update step** (line 5). We update $\pi^{(t+1)}(i,j) = \max\{0, \, \pi^{(t)}(i,j) - \eta \cdot \nabla_{\pi(i,j)} F\}$, which is a simple arithmetic operation plus comparison, repeated $m \times n$ times. Thus, $O(mn)$ operations.

- **Projection onto feasible set** (line 6). For each $i$, we normalize $\{\pi^{(t+1)}(i,j)\}_{j=1}^n$ by dividing each entry by the sum $\sum_{j=1}^n \pi^{(t+1)}(i,j)$. Computing this sum and the subsequent division also requires $O(mn)$ time in total.

- **Convergence check** (line 7). Computing the Frobenius norm $\|\pi^{(t+1)} - \pi^{(t)}\|_F$ requires $O(mn)$ operations.

Since all five steps above are each $O(mn)$ per iteration, the total cost per iteration is $O(mn)$. Over $T$ iterations, the overall complexity becomes

$$O(mn) \times T = O(mnT).$$

Hence, the time complexity of the algorithm is $O(mnT)$.

### D.6   Proof of Prop. 5

This proposition follows directly from the proposition presented in Zaev [2015].

**Proposition**   With conditions of Theorem 1, the OT problem with linear constraints has a solution if and only if the set $\Gamma_{\mathcal{W}} := \{\pi \in \Gamma : \int w d\pi = 0, w \in \overline{\mathcal{W}}\}$ is not empty, where $\overline{\mathcal{W}}$, the closure of $\mathcal{W}$ in the $C_L$ topology.

To apply the proposition, we need to verify that its assumptions are satisfied in the $C_L$ topology. Since the functions are continuous and have compact support, they clearly meet the required conditions. Thus, we can use the proposition to complete the proof.

## E   Computational Experiments Supplementary Materials

### E.1   Projected-Gradient CCE Solver

---

**Algorithm 1** Projected-Gradient CCE Solver

---

**Require:** Feature sets $X = \{x_i\}_{i=1}^N$, labels $Y = \{y_j\}_{j=1}^N$, probabilities $\mathbb{P}$, distributions $X^- = \{x_i : y_i = -1\}$, $X^+ = \{x_j : y_j = +1\}$, cost matrix $C = \{c_{ij}\}$ with $c_{ij} = c(X_i, X_j)$, regularization parameters $\lambda_1, \lambda_2$, step size $\eta$, threshold $\epsilon$, and max iterations $T$.

1: **Initialize:** $\pi^{(0)}(i,j) \geq 0$ for all $i, j$ (e.g., uniform).
2: **for** $t = 0$ to $T-1$ **do**
3:     **Compute marginals**:

$$\pi_1^{(t)}(i) = \sum_j \pi^{(t)}(i,j), \quad \pi_2^{(t)}(j) = \sum_i \pi^{(t)}(i,j).$$

4:     **Compute gradient**:

$$\nabla_{\pi(i,j)} F = c_{ij} + \lambda_1 \left( \ln\left(\frac{\pi_1^{(t)}(i)}{\mathbb{P}_-(i)}\right) + 1 \right) + 2\lambda_2 \frac{\pi_2^{(t)}(j) - \mathbb{P}_+(j)}{\mathbb{P}_+(j)}.$$

5:     **Gradient step with positivity projection**:

$$\tilde{\pi}^{(t+1)}(i,j) = \max\left\{ 0, \pi^{(t)}(i,j) - \eta \nabla_{\pi(i,j)} F \right\}.$$

6:     **Update plan**:

$$\pi^{(t+1)} \leftarrow \tilde{\pi}^{(t+1)}.$$

7:     **Check convergence**:

$$\textbf{if } \left\| \pi^{(t+1)} - \pi^{(t)} \right\|_F < \epsilon, \quad \textbf{terminate}.$$

8: **end for**
9: **Return** $\pi^{(T)}$.

---

### E.2   Unbalanced Sinkhorn's algorithm

The unbalanced Sinkhorn algorithm (Peyré et al., 2017, § 10) is an extension of the classic Sinkhorn algorithm, adapted for solving optimal transport problems where the mass of the distributions does not necessarily match. The Unbalanced Sinkhorn algorithm modifies this problem to allow for

differences in mass between $\mathbf{P}$ and $\mathbf{Q}$. The constraints are relaxed using so-called Kullback-Leibler (KL) divergence terms, leading to the unbalanced optimal transport problem:

$$\min_{\mathbf{T} \geq 0} \langle \mathbf{T}, \mathbf{C} \rangle_F - \epsilon \cdot H(\mathbf{T}) + \lambda_1 \cdot \text{KL}(\mathbf{T}\mathbf{1}_M \| \mathbf{P}) + \lambda_2 \cdot \text{KL}(\mathbf{T}^\top \mathbf{1}_N \| \mathbf{Q})$$

Here, $\lambda_1$ and $\lambda_2$ are regularization parameters for the marginal constraints, and the KL divergence terms $\text{KL}(\cdot \| \cdot)$ measure the discrepancy between the marginals of the transport plan $\mathbf{T}$ and the given distributions

In empirical settings, the algorithm deals with discrete distributions often derived from data samples. This involves computing a transport plan between empirical distributions, which are represented as sums of Dirac masses. The empirical part of the algorithm refers to its application to empirical distributions, i.e., distributions represented by samples (data points), which is a common scenario in practical applications.

---

**Algorithm 2** Unbalanced Sinkhorn Optimal Transport

---

**Input:** probability measures $P = (p_i)_i \in \mathbb{R}^n$ and $Q = (q_j)_j \in \mathbb{R}^m$, cost matrix $C = (c_{ij})_{ij} \in \mathbb{R}^{n \times m}$, regularization parameter $\epsilon$, $\lambda_1$ and $\lambda_2$ regularization parameters and number of iterations $N$;

**Output:** approximated optimal transport matrix $\pi$;

**Initialize:** $u_0 = \mathbf{1}_n$, $v_0 = \mathbf{1}_m$;

**Compute:** $K = e^{-\epsilon C}$

**for** $n = 0$ **to** $N - 1$ **do**

   Update $u_{n+1} = \left( \frac{P}{Kv_n} \right)^{\frac{\lambda_1}{\epsilon + \lambda_1}}$

   Update $v_{n+1} = \left( \frac{Q}{K^\top u_n} \right)^{\frac{\lambda_2}{\epsilon + \lambda_2}}$

**end for**

**Return:** $\mathbf{T} = \text{diag}(u_N) K \text{diag}(v_N)$;

---

### E.3 The back-and-forth method

The Back-and-Forth Jacobs and Léger [2020] method offers a robust solution for computing optimal transport maps with strictly convex costs, including p-power costs, for probability densities $\mathbb{P}$ and $\mathbb{Q}$ on an $n$-point grid. This method, characterized by its computational efficiency, requires $O(n)$ storage and $O(n \log(n))$ computation per iteration. The iteration count needed to achieve $\epsilon$ accuracy is proportional to $O(\max(\|P\|_\infty, \|Q\|_\infty) \log(\frac{1}{\epsilon}))$, showcasing the method's effectiveness in both storage and computational resource optimization.

In the back-and-forth method, $\Omega$ is considered as a convex and compact subset of $\mathbb{R}^d$ and focuses on a specific cost function $c \colon \Omega \times \Omega \to \mathbb{R}$ defined as $c(x, y) = h(y - x)$. Here, $h \colon \mathbb{R}^d \to \mathbb{R}$ is a strictly convex and even function. The dual Kantorovich problem is considered in the two following equivalent forms:

$$I(\psi) = \int \psi \, d\mu + \int \psi^c \, d\nu \quad \text{or} \quad J(\phi) = \int \varphi \, d\nu + \int \varphi^c \, d\mu,$$

where $\varphi^c, \psi^c$ are $c$-transformation of $\varphi, \psi$. The gradient of $J(\phi)$ in the space of functions from $\Omega$ to $\mathbb{R}$ can be written as ( Jacobs and Léger, 2020, Lemma 3):

$$\nabla J(\phi) = (-\Delta)^{-1} (\nu - T_{\varphi *} P),$$

where the $\Delta$ is Laplacian operator and

$$T_\varphi(x) = x - (\nabla h)^{-1} (\nabla \varphi^c(x))$$

is the explicit solution of c-transformation (Jacobs and Léger, 2020, Lemma 1). We now introduce the gradient descent back-and-forth method.

Theorem establishes that if $\varphi$ is the solution to the dual Kantorovich problem, then $T_\varphi$ constitutes the optimal map.

**Algorithm 3 Back-and-Forth** Method for Optimal Map
___
**Input:** probability measures $\mathbb{P}$ and $\mathbb{Q}$, cost function $c$ and number of iterations $N$;
**Output:** approximated Kantorovich potential functions;
**Initialize:** set $\varphi_0 = 0, \psi_0 = 0$
**for** $n = 0$ **to** $N - 1$ **do**
$\quad \varphi_{n+\frac{1}{2}} = \varphi_n + \sigma \nabla J(\varphi_n),$
$\quad \psi_{n+\frac{1}{2}} = (\varphi_{n+\frac{1}{2}})^c,$
$\quad \psi_{n+1} = \psi_{n+\frac{1}{2}} + \sigma \nabla I(\psi_{n+\frac{1}{2}}),$
$\quad \varphi_{n+1} = (\psi_{n+1})^c.$
**end for**
**Return:** $\varphi_N, \psi_N$;
___

### E.4 Entropy-based Algorithm for Collective Counterfactual Explanations

The algorithm provides an entropy-regularized solution for generating collective counterfactual explanations by optimizing a transport plan $\pi$ that balances transportation cost, adherence to given probability distributions, and entropy maximization. Starting with initialized potentials, the algorithm iteratively refines $\pi$ using gradient-based updates, ensuring constraints on marginals and regularization terms such as Kullback-Leibler divergence, $\chi^2$-divergence, and entropy thresholds. Convergence is determined by the Frobenius norm of consecutive transport plans, offering a robust and interpretable framework for generating collective CE.

## F   Supplementary Simulation

In our numerical study, we selected two actionable features from each dataset to evaluate the effectiveness of our algorithmic recourse method. For the **Adult** dataset, we used *Education Level* and *Hours per Week*, which are relevant for assessing socioeconomic mobility. From the **COMPAS** dataset, we chose *Priors Count* and *Length of Stay*, which capture key aspects of criminal history and detention. For the **HELOC** dataset, *Percent Trades* and *Trades Number* were selected, reflecting financial behavior and creditworthiness. Finally, for the synthetic **Moons** dataset, we utilized *Feature* from both dimensions to simulate simple, interpretable feature changes.

### F.1   Hyperparameters of Different Methods

This section presents the hyperparameters used for various methods in a compact format.

| Method | Hyperparameters |
|---|---|
| Wachter | `loss_type=BCE, t_max_min=1/60` |
| Roar | `lr=0.01, lambda_=0.01, delta_max=0.001, t_max_min=0.5, loss_type=BCE, y_target=[0,1], loss_threshold=1e-3, discretize=False, sample=True` |
| CCHVAE | `n_search_samples=100, p_norm=2, step=1e-2, max_iter=1000, clamp=True, binary_cat_features=True,` **VAE:** `layers=[|features| - |immutables|, 256, 2], train=True, lambda_reg=1e-6, epochs=500, lr=1e-3, batch_size=32` |
| Growing Spheres | No hyperparameters specified |
| FOCUS | `optimizer=adam, lr=0.001, n_class=2, n_iter=1000, sigma=1.0, temperature=1.0, distance_weight=0.01, distance_func=l1` |
| CLUE | `train_vae=True, width=10, depth=5, latent_dim=12, batch_size=20, epochs=5, lr=0.001, early_stop=20` |

Table 1: Hyperparameters for different methods used in experiments in the Carla Package.

**Algorithm 4** Entropy-based Solution for Collective Counterfactual Explanations

**Require:**

- Feature sets $X = \{x_i\}_{i=1}^N$, $Y = \{y_j\}_{j=1}^N$
- Probability distributions $\mathbb{P}_-$, $\mathbb{P}_+$
- Cost matrix $C = \{c_{ij}\}$
- Regularization parameters $\lambda_1, \lambda_2, \gamma, \epsilon_\pi$
- Step size $\eta$, tolerance $\epsilon$, max iterations $T$

1: **Objective Function**:

$$F(\pi) = \sum_{i,j} \pi_{ij}\, c_{ij} + \lambda_1\, D_{\mathrm{KL}}\big(\pi_1 \,\|\, \mathbb{P}_-\big) + \lambda_2\, D_{\chi^2}\big(\pi_2 \,\|\, \mathbb{P}_+\big) + \epsilon_\pi \sum_{i,j} \pi_{ij}\, \log\big(\pi_{ij}\big),$$

where $\pi_1(i) = \sum_j \pi_{ij}$, $\pi_2(j) = \sum_i \pi_{ij}$.

2: **Parameterization**:

$$\pi_{ij} = \exp\Big( \tfrac{\phi_i + \psi_j - c_{ij}}{\gamma} \Big).$$

3: **Initialize**: potentials $\phi^{(0)} \in \mathbb{R}^N$, $\psi^{(0)} \in \mathbb{R}^N$ (e.g., zero vectors).

4: **for** $t = 0$ to $T - 1$ **do**

5:     **Compute current transport plan**:

$$\pi_{ij}^{(t)} = \exp\Big( \tfrac{\phi_i^{(t)} + \psi_j^{(t)} - c_{ij}}{\gamma} \Big).$$

6:     **Compute marginals**:

$$\pi_1^{(t)}(i) = \sum_j \pi_{ij}^{(t)}, \qquad \pi_2^{(t)}(j) = \sum_i \pi_{ij}^{(t)}.$$

7:     **Compute gradients via chain rule**:

$$\nabla_{\phi_i} F = \frac{1}{\gamma} \sum_j \pi_{ij}^{(t)} \Big[ c_{ij} + \lambda_1 \Big( \log\big( \tfrac{\pi_1^{(t)}(i)}{\mathbb{P}_-(i)} \big) + 1 \Big) + 2\,\lambda_2\, \frac{\pi_2^{(t)}(j) - \mathbb{P}_+(j)}{\mathbb{P}_+(j)} + \epsilon_\pi \Big( \log(\pi_{ij}^{(t)}) + 1 \Big) \Big].$$

$$\nabla_{\psi_j} F = \frac{1}{\gamma} \sum_i \pi_{ij}^{(t)} \Big[ c_{ij} + \lambda_1 \Big( \log\big( \tfrac{\pi_1^{(t)}(i)}{\mathbb{P}_-(i)} \big) + 1 \Big) + 2\,\lambda_2\, \frac{\pi_2^{(t)}(j) - \mathbb{P}_+(j)}{\mathbb{P}_+(j)} + \epsilon_\pi \Big( \log(\pi_{ij}^{(t)}) + 1 \Big) \Big].$$

8:     **Gradient update**:

$$\phi_i^{(t+1)} = \phi_i^{(t)} - \eta\, \nabla_{\phi_i} F, \qquad \psi_j^{(t+1)} = \psi_j^{(t)} - \eta\, \nabla_{\psi_j} F.$$

9:     **Check convergence** (e.g., via plan difference or potential difference):

$$\textbf{if } \big\| \pi^{(t+1)} - \pi^{(t)} \big\|_F < \epsilon \textbf{ then terminate.}$$

10: **end for**

11: **Return** final plan $\pi^{(T)} = \exp\Big( \tfrac{\phi^{(T)} + \psi^{(T)} - C}{\gamma} \Big)$.

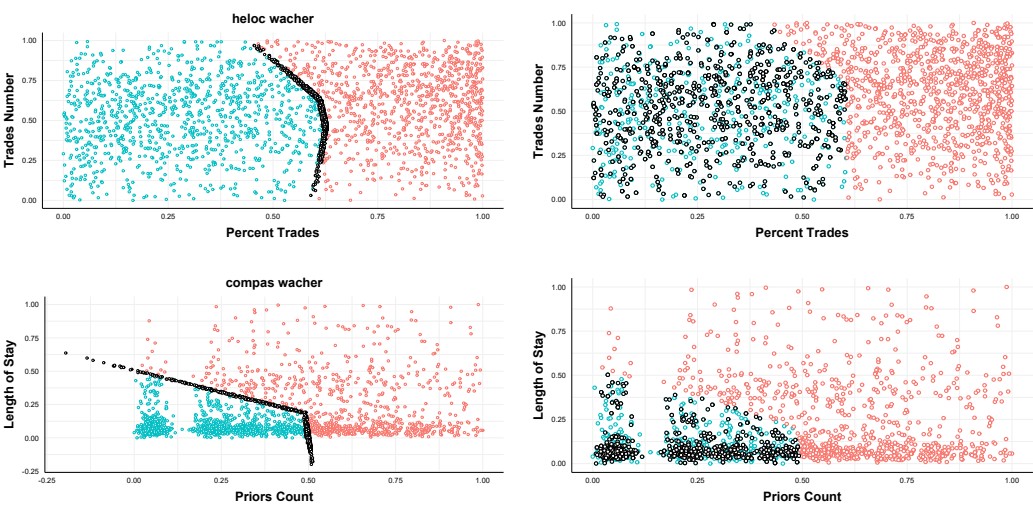

Figure 6: Top left: HELOC dataset with its Wachter counterfactual explanations. Top right: Collective counterfactual explanations. Bottom left and bottom right: Wachter and collective counterfactual explanations, respectively.

