# OpenReview forum: "Collective Counterfactual Explanations: Balancing Individual Goals and Collective Dynamics"
_NeurIPS.cc/2025/Conference — NeurIPS 2025 poster_

### Official Review · Reviewer_fXWP · 2025-06-16

**Clarity:** 3
**Significance:** 3
**Originality:** 3
**Rating:** 5
**Confidence:** 4

**Summary:**

This paper introduces a new framework for group-wise counterfactual explanation, named collective counterfactual explanation, that aims to maintain the distribution of perturbed instances and positive instances as unchanged as possible while minimizing the total modification costs.
The authors show that this task can be reduced to the optimal transport problem, which can be efficiently solved by existing algorithms such as the Sinkhorn algorithm.
Experimental results demonstrated that the proposed method could achieve significantly lower competition costs than existing standard methods and balance the trade-off between competition cost and modification cost by tuning the hyperparameter.

**Questions:**

1. In experiments, the authors state, "We use only two continuous actionable features to determine the best recourse." What is the reason for this setting? Can the proposed method be applied to situations with more than two actionable features?
1. How were the average running times of the proposed method compared to the other baseline methods?
1. (typos) In line 28, "desirable" and "undesirable" may be reversed.

**Ethical Concerns:**

["NO or VERY MINOR ethics concerns only"]

**Final Justification:**

Thank you for taking the time to answer all of my questions and concerns. Since the rebuttal addressed my concerns in part, I have decided to keep my score positive.

**Limitations:**

The authors discuss the limitations of their work, but there is no discussion of its potential negative societal impacts.

**Paper Formatting Concerns:**

I could not find any formatting issues in this paper.

**Quality:**

3

**Strengths And Weaknesses:**

Strengths
1. This paper is well-written and well-organized.
1. The authors formulate a new, well-motivated problem of group-wise counterfactual explanation. I find it interesting to solve the formulated problem by leveraging techniques from optimal transport.
1. The efficacy of the proposed method is well-supported by the experimental results.

Weaknesses
1. One of my concerns is that counterfactuals are limited to a predefined finite set of positive samples $\mathcal{X}^+$, in contrast to the existing standard methods that optimize counterfactuals over an infinite set of perturbations. I think it is challenging for the proposed method to meet some of the desiderata of counterfactual explanation, such as sparsity.
1. Another concern is the computational cost of the proposed method, as the experiments are limited to a setting where only two continuous features can be modified, and the average running time of the proposed method is not reported.

---

> ### Author Rebuttal · Authors · 2025-07-31
>
> We thank the reviewer for their insightful review. We're pleased that they found the paper well-written and well-organized, the problem well-motivated, our optimal transport framing interesting, and the proposed method efficient and well-supported by experiment. Below, we address the reviewer’s questions and concerns.
>
> > Re. I think it is challenging for the proposed method to meet some of the desiderata of counterfactual explanation, such as sparsity.
>
> We thank the reviewer for the clarifying question. To clarify, $\mathcal{X}^+$ is defined by the classifier and can be a continuous set. In this sense, we are not losing generality compared to standard methods. However, we acknowledge that the available positive samples may sometimes be insufficient to define a reliable support for transport. In such cases, it is possible to augment the data—e.g., by generating synthetic points or using bootstrapping—to ensure a more representative distribution of both negative and positive instances.
>
>
> > Re. Another concern is the computational cost of the proposed method, as the experiments are limited to a setting where only two continuous features can be modified ... Can the proposed method be applied to situations with more than two actionable features?
>
> We thank the reviewer for raising this important point. In many benchmark settings—including those established in the CARLA library, which is widely adopted in the recourse literature—it is common practice to restrict evaluation to 2–3 continuous actionable features. This choice is also made in prior works such as Pawelczyk et al. (2020, C-CHVAE) and Mothilal et al. (2020, DiCE). **We have followed this convention, though our method can be applied to a larger set of features**. Adhering to this setup also allowed us to visually illustrate how CCE distributes individuals under collective dynamics.
>
> Regarding computational cost, **Proposition 4** establishes that our unbalanced optimal transport (UOT) solver has a **theoretical time complexity of $O(n^2)$**, where $n$ is the number of instances. Thus, the method scales reasonably in practice and does not suffer from the curse of dimensionality in its algorithmic formulation. However, we acknowledge that empirical distributions in high-dimensional spaces are inherently sparse, which can affect the stability of the transport plan. In such cases, optimal literature rich literature offers different solutions: For example, one can impose smoothness assumptions on the underlying data distribution like as *smoothed optimal transport* or leverage techniques such as *sliced optimal transport*, which project high-dimensional data into one-dimensional subspaces and significantly reduce computational complexity while preserving key geometric structure
>
>
> > Re. How were the average running times of the proposed method compared to the other baseline methods?
>
>
> While we did not report average running times in the current submission, the proposed method is based on unbalanced optimal transport with **$O(n^2)$** complexity (see Proposition 4), and it typically runs in **seconds** for standard benchmark settings. While we cannot make precise complexity claims about the implementations of baseline methods in the CARLA package---since their internal optimizations and dependencies may vary---we provide empirical runtime comparisons for the Moons dataset to offer practical insight.
>
> The table below shows the time (in seconds) required to generate **1,000 counterfactuals** on the Moons dataset. These experiments were run on a standard personal laptop a single CPU and 16 GB RAM.
>
> | Method           | Time (s) |
> | ---------------- | -------- |
> | wacher           | 6.38   |
> | growing\_spheres | 10.82  |
> | cce (ours)       | 15.64  |
> | focus            | 39.66  |
> | cchvae           | 90.18  |
> | cchvae           | 147.56 |
>
> Note that runtime can vary across datasets and methods, particularly for approaches like C-CHVAE that rely on generative models and are more sensitive to the number of features. Nevertheless, **CCE demonstrates competitive runtime performance**, even without GPU acceleration, and scales well for practical applications.

---

> > ### Comment · Reviewer_fXWP · 2025-08-02
> >
> > Thank you for taking the time to answer all of my questions and concerns. Since the rebuttal addressed my concerns in part, I have decided to keep my score positive.

---

> > > ### Author Response · Authors · 2025-08-05
> > >
> > > Thank you very much for your thoughtful comments and for considering our rebuttal carefully. We truly appreciate your positive evaluation. If any specific concerns remain that we can further address, please let us know—we would be more than happy to clarify further.

---

### Official Review · Reviewer_Aeab · 2025-06-29

**Clarity:** 3
**Significance:** 3
**Originality:** 3
**Rating:** 5
**Confidence:** 2

**Summary:**

Most previous works on counterfactual explanations (CE) focus only on individual cases and overlook the externalities that occur when many people follow similar recommendations. This work proposes a new framework to address this limitation. Building on a population dynamics model, the authors introduce a penalty term into the standard CE objective to balance private and external costs and develop a scalable algorithm to solve it efficiently. In addition, they analyze how this approach improves over prior methods across different criteria for successful recourse and demonstrate its versatility by extending it to path-guided CE and ordered classifier families.

**Questions:**

1. Why use KL-divergence for the regularization term corresponding to $\lambda_1$?
2. What happens if the two regularization terms in Equation (6) use the same or different divergences? How does this affect choosing $\lambda_1$ and $\lambda_2$?
3. Is using a discretized grid a good way to compute $\chi^2$-divergence? It seems hard to control the grid resolution, and this might strongly affect the results.

**Ethical Concerns:**

["NO or VERY MINOR ethics concerns only"]

**Final Justification:**

Since most of my concerns are addressed, I remain supportive of acceptance.

**Limitations:**

Yes

**Paper Formatting Concerns:**

No major formatting issues

**Quality:**

3

**Strengths And Weaknesses:**

I recommend accepting this paper based on the following strengths:

- On the topic of accounting for externalities, prior work [56] was the first to highlight the existence of this problem, but it only proposed some ad-hoc mitigation strategies. To the best of my knowledge, this work is the first to provide a complete and practical framework as a solution. For this reason, it is likely to serve as a solid benchmark for future research in this area. I also look forward to the authors releasing the source code.
- The writing quality of this paper is good. The logic is clear and rigorous, and the authors have helpfully organized the necessary background material in Appendices A–C for better understanding. I briefly reviewed the proofs in Appendix D and did not notice any major issues. Additionally, the appendices detailedly provide the algorithms and experimental, ensuring the reproducibility of this work.
- The experimental results strongly support the authors’ arguments. For example, Figures 1, 5b, and 6 demonstrate that the proposed method indeed produces actionable recommendations and that the individuals receiving recourse end up closer to a realistic distribution. Meanwhile, Figure 4 confirms that the approach balances private and external costs as expected.

In addition to these strengths, I would like to point out a few weaknesses that the authors could improve:

- I would appreciate an explanation of why the regularization term corresponding to $\lambda_1$ uses KL-divergence. Using different divergence measures for the two regularization terms could potentially introduce challenges, such as making the choice of $\lambda_2$ more sensitive.
- The assumption that “individuals are in equilibrium before CE” is an important premise in this paper. I suggest that the authors explicitly state this assumption in a formal way.

Note that I didn't pay much attention to this field in the past, so I gave a relatively low confidence.

---

> ### Author Rebuttal · Authors · 2025-07-31
>
> We sincerely thank the reviewer for the insightful and encouraging comments. We are heartwarmed that the reviewer found our approach novel and impactful, and we appreciate their close reading of both the theoretical and experimental components. Below we discuss their thoughtful suggestions and questions in detail.
>
> > Re. Why use KL-divergence for the regularization term corresponding to $\lambda_1$?
>
> We use different divergence measures for the two regularization terms in Eq. (6) to take advantage of their respective computational and modeling properties:
>
> * For the divergence between the *second marginal* and the equilibrium distribution $D_{\chi^2}(\pi_2 \,\|\, P^+)$, we use the $\chi^2$-divergence because it naturally aligns with our equilibrium penalty, which quantifies deviations from the target equilibrium distribution (see Proposition 1 and Section 2.2). Specifically,
>   $$
>   D_{\chi^2}(P^{\text{CE}} \,\|\, P^+) = \int \left( \frac{dP^{\text{CE}}}{dP^+} - 1 \right)^2 dP^+,
>   $$
>   provides a tractable and interpretable penalty that reflects the degree of over- or under-allocation of individuals relative to available resources. This choice is particularly suitable because it penalizes both *overcrowding* and *underutilization* in a simple quadratic form—exactly the kind of behavior we want to discourage in population-level optimization. While we could use KL-divergence instead of $D_{\chi^2}$, the resulting penalty would be asymmetric: it more strongly penalizes cases where $\frac{dP}{dQ}$ is large (i.e., overcrowding) than when $\frac{dP}{dQ} \to 0$ (i.e., underutilization). We found $\chi^2$-divergence to better align with our intuition that deviations from equilibrium should be penalized symmetrically. Of course, this is a design choice, and we will clarify our reasoning in the revision.
>
>
> * For the regularization over the *first marginal* $D_{\text{KL}}(\pi_1 \,\|\, P^-)$, we adopt KL-divergence due to the popularity of entropy-regularized optimal transport and its well-known properties. This choice enables the use of scalable solvers inspired by Sinkhorn-style algorithms and ensures the relaxed optimization in Eq. (6) remains strongly convex under mild assumptions.
>
> > Re. Using different divergence measures for the two regularization terms could potentially introduce challenges ... What happens if the two regularization terms in Equation (6) use the same or different divergences? How does this affect choosing $\lambda_1$ and $\lambda_2$?
>
> * We believe that using different divergences does not compromise the solution’s stability, and our experiments show no indication of such issues. In fact, the KL term $\lambda_1 D_{\text{KL}}(\pi_1 ,|, P^-)$ acts as a soft constraint, and Proposition 3 shows that as $\lambda_1 \to \infty$, the relaxed solution converges to the exact one, i.e., $\pi^{(\lambda_1)} \to \pi^*$. Thus, we generally expect this divergence term to provide both computational flexibility and theoretical guarantees without introducing any difficulty. We appreciate the reviewer’s careful attention to this detail.
>
> * Regarding the choice of hyperparameters when the divergences are the same: $\lambda_1$ is primarily an optimization parameter, controlling the trade-off between enforcing a hard constraint and easing optimization. On the other hand, $\lambda_2$ reflects the policy maker’s trade-off between individual modification cost and collective competition. A higher $\lambda_2$ prioritizes population-level balance, reducing overcrowding but allowing higher individual effort, while a lower value favors minimal recourse cost at the risk of systemic congestion. Therefore, regardless of the specific divergence used (for which we recommend $\chi^2$), $\lambda_2$ should be tuned based on the desired balance between social externalities and individual burden in the given application.
>
> > Re. I suggest that the authors explicitly state [the equilibrium] assumption in a formal way.
>
> This is a great suggestion, and we will ensure the equilibrium assumption is clearly highlighted in our updated manuscript. That being said, our framework can be extended to non-equilibrium settings as we have discussed in response to Reviewer J14q.
>
> > Re. Is using a discretized grid a good way to compute $\chi^2$-divergence?
>
> We appreciate this insightful question. Discretizing the feature space into a grid is a practical approximation for computing $D_{\chi^2}(P^{\mathrm{CE}} \| P^+)$, especially in low-dimensional settings such as the Moons dataset or when interpretability is important. In our simulations, the number of actionable features is fewer than three, making grid-based estimation both feasible and stable. While grid resolution can influence the computed values, we ensure consistency across all methods by using a fixed grid and validating results through convergence checks. For higher-dimensional or more sensitive scenarios, we agree that adaptive approaches—such as kernel density estimation or nearest-neighbor smoothing—could offer improved robustness and are a promising direction for future refinement.

---

> > ### Comment · Reviewer_Aeab · 2025-08-05
> >
> > Thank you for the detailed clarifications. I have no further concerns and do not wish to revise my score. I look forward to seeing these clarifications and insights reflected in the revised version of the paper.

---

> > > ### Author Response · Authors · 2025-08-05
> > >
> > > Thank you very much for your thoughtful engagement and positive feedback. We will make sure your valuable suggestions and clarifications are reflected in the revised manuscript.

---

### Official Review · Reviewer_kVUx · 2025-07-02

**Clarity:** 3
**Significance:** 2
**Originality:** 4
**Rating:** 4
**Confidence:** 4

**Summary:**

This work introduces a new formulation of counterfactual explanations generation which targets to evenly distribute the counterfactual points of negatively affected individuals within the data distribution of the favourable region. The motivation behind this approach is to mitigate the potential competition for limited resources after counterfactuals are implemented, especially if the counterfactuals mostly converge to similar spaces in the favourable region. The method is evaluated theoretically and experimentally. The evaluation involves 3 tabular datasets, 2 types of machine learning models (MLP and random forest), 6 baseline methods (2 of which target plausibility of counterfactuals). Results on the cost of competition are promising, but recourse cost (distance between counterfactuals and inputs) for the proposed method is not the best.

**Questions:**

- Could you comment on how different your targeted problem setting is to the plausibility problem, i.e., requiring the generated explanations to be located well within the data manifold?
- It seems to me intuitively that existing methods targeting plausibility and diversity, like DiCE or MOC, can naturally address the collective dynamics problem. Do you have any insights on that?
- Minor point: there's a typo in 28 regarding desirable vs. undesirable; there's a dot above line 291 which takes some spaces.

Also see comments on weaknesses.

**Ethical Concerns:**

["NO or VERY MINOR ethics concerns only"]

**Final Justification:**

The paper is sound in terms of methodology. The rebuttal with additional baselines addressed my main concerns about the significance of results. The targeted problem is quite artificial for the counterfactual explanations pipeline, which could be a fundamental limitation of this work.

**Limitations:**

It would be important to address specific differences between the targeted scenario and the plausibility problem.

**Quality:**

3

**Strengths And Weaknesses:**

Strengths:
- The paper is very complete and well-written with clear motivations, illustrations, assumptions, etc.
- The mathematical formulations of the problem and the methodology are clear and easy to follow. Theoretical analyses are also adequate.
- While the problem setting targeted in this paper seems very similar to the plausibility of counterfactuals (requiring the generated explanations to be located well within the data manifold), the characterisations regarding resources and equilibrium are novel.

Weaknesses:
- The main concern I have is the validity and significance of experimental results. Given the similarity between the problem setting and the plausibility of counterfactuals, it would make sense to compare the proposed method with existing methods which also target plausibility. There are 6 baselines involved in the experiment, but only 2 of them (CLUE and CCHVAE) consider the plausibility aspect. Those two methods both require, and are quite sensitive to the quality of, some external generative models. There exist many other methods that might perform better in terms of plausibility, therefore potentially better cost of competition. For example, FACE, DiCE (these two are also in the CARLA library), Guided Prototypes, MOC. All names are taken from Table 1, page 5 of Karimi et al.'s ACM Computing Survey (https://dl.acm.org/doi/pdf/10.1145/3527848). Personally, I've observed much better plausibility results from FACE and DiCE than CCHVAE. The presented good performance on cost of competition also comes with significant trade-offs with recourse cost, especially given that the baselines do not give the best recourse cost in the literature - those by mixed integer programming (e.g., Mohammadi et al. AIES 2021, https://dl.acm.org/doi/10.1145/3461702.3462514) usually perform better. I think addressing this point will make the paper a much stronger one.
- In addition to the presented evaluation metrics, including quantitative measures for plausibility and computation time would be desirable.
- Clarity on related works: following the above points, it'd be clearer if the characteristics, especially which desirable properties the baselines consider, are explicitly discussed.

---

> ### Author Rebuttal · Authors · 2025-07-31
>
> We thank the reviewer for the insightful review of our work. We are glad they found the paper complete and well-written, with clear motivations, illustrations, and assumptions, as well as adequate theoretical analysis and a novel characterization of resources and equilibrium. Below, we address the reviewer’s specific concerns and questions.
>
>
> > Re. Could you comment on how different your targeted problem setting is to the plausibility problem, i.e., requiring the generated explanations to be located well within the data manifold?
>
> Generally, plausibility-focused methods like DiCE, MOC, or Guided Prototypes focus on **individual** feasibility, plausibility, or sparsity but not on the externalities that arise when many individuals receive similar recommendations. In contrast, our framework directly incorporates collective impact of recommendations in addition to individual costs. This collective optimization perspective sets our method apart.
>
> To illustrate this distinction more clearly: in plausibility-focused methods, it's possible for all individuals to be recommended the same counterfactual point, as long as it lies on the data manifold and the individual cost is low. However, in real-world scenarios, this leads to overcrowding, which traditional methods fail to anticipate. Our approach asserts that if a significant number of individuals are advised to move toward the same point, **externalities will arise** due to natural resource constraints, thus, any responsible recourse method should incorporate such effects.
>
> > Re. ... it would make sense to compare the proposed method with existing methods which also target plausibility. There are 6 baselines involved in the experiment, but only 2 of them (CLUE and CCHVAE) consider the plausibility aspect. ... There exist many other methods that might perform better in terms of plausibility, therefore potentially better cost of competition. For example, FACE, DiCE, Guided Prototypes, MOC.
>
> We thank the reviewer for this suggestion. To further enrich our comparisons and better support our claims empirically, we have conducted additional experiments based on the reviewer’s suggestions, comparing our method (CCE) with four plausibility-oriented baselines: **FACE, DiCE, Guided Prototypes, and MOC**. Table 1 and 2 below report these new results.
>
> As expected, since these plausibility-based methods impose only local constraints on individual recourse, congestion arises when a large share of the population seeks recourse, leading to significantly **higher competition costs**. Our method, by design, mitigates this effect.
>
> #### Table 1: Competition Cost Comparison
> Below is a comparison of the **competition cost** across methods:
>
> | AR       | Moons      | HELOC      | COMPAS     | Adult      |
> | -------- | ---------- | ---------- | ---------- | ---------- |
> | **CCE** | **0.0599** | **0.0633** | **0.0796** | **0.0688** |
> | DICE     | 0.3961     | 0.1760     | 0.3423     | 0.3749     |
> | FACE     | 0.6700     | 0.3560     | 0.2523     | 0.6753     |
> | Guided Prototypes       | 0.8361     | 0.7959     | 0.7336     | 0.6118     |
> | MOC      | 0.4827     | 0.4613     | 0.5135     | 0.2996     |
>
> *Mean values of competition cost (lower is better).*
>
> Considering the **total cost** (modification + competition), our method consistently outperforms the baselines. This is because **CCE uniquely and significantly addresses the cost of competition**.
>
> #### Table 2: Total Cost Comparison
>
> | AR       | Moons      | HELOC      | COMPAS     | Adult      |
> | -------- | ---------- | ---------- | ---------- | ---------- |
> | **CCE** | **0.7127** | **0.8271** | **0.7629** | **0.5579** |
> | DICE     | 1.7231     | 1.1252     | 1.4965     | 1.5560     |
> | FACE     | 2.1115     | 1.1892     | 0.8661     | 2.0792     |
> | Guided Prototypes       | 2.7834     | 2.6358     | 2.4119     | 1.9578     |
> | MOC      | 1.8392     | 1.7983     | 1.9086     | 1.2055     |
>
> *Mean values of total cost (lower is better).*
>
>
> > Re. It seems to me intuitively that existing methods targeting plausibility and diversity, like DiCE or MOC, can naturally address the collective dynamics problem. Do you have any insights on that?
>
> While related, individual plausibility does not necessarily address competition costs, for the reasons discussed above. In prior work, plausibility is often equated with closeness to the data manifold. These approaches focus on individual recourse and typically overlook the *distributional* properties of the manifold, which are crucial when many individuals seek similar recourse and may crowd the same regions. In contrast, our work explicitly addresses *distribution-aware* recourse. By incorporating equilibrium population density into the objective, we offer a principled mechanism for managing collective dynamics and mitigating congestion.
>
>
> > Re. computation time
>
> Regarding computation time, we provide a theoretical analysis in Proposition 4, which characterizes the time complexity of our gradient-based unbalanced optimal transport solver as $O(Tmn)$, where $T$ is the number of iterations, and $m, n$ are the sizes of the source and target distributions, respectively. This offers a practical estimate of scalability and efficiency. We also report the computation time of our algorithm alongside other baselines, as run on a local machine with a single CPU, in our response to Reviewer fXWP below.
>
> Last but not least, we thank the reviewer for their helpful suggestions to explicitly report desirable properties of the baselines and include additional plausibility metrics. We also thank them for spotting a typo. We will incorporate these in our revision.

---

> > ### Comment · Reviewer_kVUx · 2025-08-05
> > **Response to author rebuttal**
> >
> > Thank you for the rebuttal. The clarification helps, and the additional comparison with new baselines makes sense. I have another question regarding the parameter $\gamma$. Could you provide some insights into how the performance will vary with different $\gamma$ values? Would the plausible counterfactual methods address the distribution problem better if $\gamma$ is very small? And what would be realistic $\gamma$ values?

---

> > > ### Author Response · Authors · 2025-08-05
> > >
> > > Thank you for the thoughtful follow-up during the rebuttal. We’re glad our clarifications resolved your concern.
> > >
> > > ---
> > > ## How performance varies with $\gamma$?
> > > We interpret $\gamma \in [0,1]$ as the fraction of negatively labeled individuals who comply with the recommended recourse (line 108). It enters the objective only through the mixture weight
> > > $$
> > > \lambda = \frac{\gamma  p_-}{\gamma  p_- + p_+},
> > > $$
> > > which defines the post–recourse distribution
> > > $$
> > > P_{\text{CE}} = \lambda  T_{\sharp} P_- + (1-\lambda)  P_+ ,
> > > $$
> > > and scales the equilibrium penalty
> > > $$
> > > D_{\chi^2} (P_{\text{CE}}  \|  P_+)
> > > = \lambda^2  D_{\chi^2} (T_{\sharp}P_-  \|  P_+).
> > > $$
> > >
> > > Therefore, by the above formulation, we can see:
> > >
> > > - As $\gamma$ increases ($\lambda\uparrow$): more mass moves toward $T_{\sharp}P_-$, so the divergence penalty grows as $\lambda^2$.
> > >   CCE reacts by shifting solutions toward the positive region and toward areas similar to the distribution of $P_{+}$ to reduce competition.
> > > - As $\gamma \to 0$ ($\lambda\to 0$): the equilibrium term vanishes and CCE reduces to standard, individual counterfactual explanations (CE).
> > >
> > > ---
> > > ## Small-$\gamma$ regime vs plausible counterfactuals.
> > > When $\gamma \approx 0$, the induced population shift is negligible ($\lambda \approx 0$), so externalities from crowding are minimal. In this regime, classical counterfactuals and CCE behave similarly because CCE’s equilibrium term is effectively off. It can easily find this point by setting $\lambda_1 = \infty$ and $\lambda_2 = 0$ in Equation 6 converting to the classical optimization problem for recourse, as explained in Proposition 1.
> > >
> > > But in practical applications, the compliance rate $\gamma$ must be non-zero, implying that at least one individual follows the recommendation. The CCE formulation, through its equilibrium penalty term $D_{\chi^{2}}(\pi_{2}||\mathbb{P}_{+})$, discourages shifting people to empty regions of the feature space. Instead, it favors states where data points from the positive class already exist, thus aligning recommendations with the current data manifold.
> > >
> > > Our simulations emphasize this point. Even with a small competition parameter, such as when $\lambda_2=0.01$, CCE still guides individuals toward higher-density regions of the positive class. This is visually demonstrated in figures like Figure 5 (left), where we have an additional total cost with respect to the classical CE. This inherent preference for plausible, data-supported recommendations is a significant advantage of the CCE framework compared to conventional CE methods, which often recommend states on the decision boundary regardless of data density.
> > >
> > > ---
> > > ## How to Determine $\gamma$ in Practice.
> > > In our framework, $\gamma$ is the expected adoption/compliance rate within the negatively labelled group $X^{-}$. Empirically, we estimate $\gamma$ from historical data—the fraction of previously negative individuals who followed through. When such data are unavailable, we use simple behavioural calibration, impose operational capacity as an upper bound, and report sensitivity over a plausible $\gamma$ range.

---

> > > > ### Comment · Reviewer_kVUx · 2025-08-07
> > > >
> > > > Thanks for the reply. The rebuttal has addressed my concerns. I am willing to raise my score to the positive side. However, I still think the problem setup, although relevant assumptions are stated, is highly artificial for the counterfactual explanations pipeline.

---

### Official Review · Reviewer_J14q · 2025-07-03

**Clarity:** 1
**Significance:** 3
**Originality:** 3
**Rating:** 2
**Confidence:** 3

**Summary:**

This paper proposes Collective Counterfactual Explanations (CCE), which reframes algorithmic recourse as a collective optimization problem. By modeling population dynamics using mean-field game theory and optimal transport, CCE accounts for negative externalities when many individuals receive similar recommendations. It balances individual costs with collective impact by penalizing deviations from population equilibrium.

**Questions:**

- How robust is the CCE method to its core "equilibrium assumption"? If the initial population distribution is not in equilibrium before CE generation, how should the framework be adjusted?

- See Weakness. Due to the paper’s lack of completeness, I believe it requires further refinement in both writing and formatting, as well as an additional round of review.

**Ethical Concerns:**

["NO or VERY MINOR ethics concerns only"]

**Final Justification:**

I appreciate the authors' response. Unfortunately, I think the core concerns I raised were not adequately addressed. Rather than responding directly to the formatting issue I pointed out, the authors shifted the discussion toward the clarity of their writing and motivation, which was not the focus of my critique.

**Limitations:**

yes

**Quality:**

1

**Strengths And Weaknesses:**

**Strengths:**
- The paper addresses a problem of significant value within the field of counterfactual inference.
- It provides solid theoretical results with proofs and demonstrates originality in its approach.

**Weaknesses:**
- My primary concern is the paper’s lack of completeness. In addition to not meeting the standard page count and lacking a dedicated conclusion section, the manuscript exhibits several formatting issues—such as a placeholder symbol (a centered dot above line 291), awkward paragraph breaks (e.g., lines 320, 348), and suboptimal paragraph structure in the final section—all of which make the paper feel unfinished.
- The core "equilibrium assumption" is not discussed in sufficient detail regarding its limitations or its feasibility in real-world tasks.
- There are multiple instances of incorrect citation formatting; the authors do not properly distinguish between \citep and \citet where appropriate (e.g., lines 230 and 232).
- While the paper presents numerous numerical studies, the code is not provided, which prevents the verification or reproduction of the experimental results. The provided GitHub link directs to a repository for a prior work, which is an unnecessary and potentially misleading inclusion.

---

> ### Author Rebuttal · Authors · 2025-07-31
>
> We thank the reviewer for reviewing our work. Below we discuss reviewer's concerns in detail.
>
> > Re. My primary concern is the paper’s lack of completeness. ... not meeting the standard page count and lacking a dedicated conclusion section, ... formatting issues
>
> With all due respect, we disagree with the reviewer’s assessment regarding the completeness of our paper. It is unclear what is meant by not meeting the standard page count or lacking a conclusion section. Our submission uses the full 9 allowed pages, with the last page fully dedicated to further situating our contributions within the broader literature, discussion, and future work, which we believe serve as a conclusion to the paper.
>
> We have also provided a comprehensive background on optimal transport in Appendices A–C, which Reviewer Aeab found “helpfully organized... for better understanding.” While we welcome concrete suggestions for improvement and thank the reviewer for pointing to the typo and wrong break, we kindly disagree that the paper is incomplete. Other reviewers have also affirmed the completeness and quality of the submission:
>
> * Reviewer kVUx: "The paper is *very complete and well‑written* with clear motivations, illustrations, assumptions, etc."
> * Reviewer fXWP: "This paper is *well-written and well-organized*."
> * Reviewer Aeab: "The writing quality of this paper is *good*. The logic is clear and rigorous, and the authors have helpfully organized the necessary background material in Appendices A–C for better understanding."
>
>
> > Re. While the paper presents numerous numerical studies, the code is not provided, which prevents the verification or reproduction of the experimental results.
>
> We have not released the code earlier solely to preserve anonymity. We have now addressed this by sharing our project on an anonymous GitHub repository. Since including links is prohibited this year, the code can be found by searching **Collective-Counterfactual-Explanations-CCE** in the GitHub search bar. Even without access to the code, we have aimed to make every aspect of our work as clear and reproducible as possible. Specifically, the appendices include pseudocode for Algorithms 1 and 4, a full listing of hyperparameters (Table 1), and detailed instructions for running each experiment. In terms of baseline, we have also replied on open-source implementation to ensure reproducibility of the comparisons.
>
> > Re. How robust is the CCE method to its core "equilibrium assumption"? If the initial population distribution is not in equilibrium before CE generation, how should the framework be adjusted?
>
> This is an intriguing question. In general, the CCE framework can be extended to non-equilibrium settings under additional assumptions, as we outline below. Recall that unlike standard methods that assume a stationary cost landscape—i.e., that the environment remains unchanged after issuing a recommendation—CCE directly models externalities resulting from recommendations. To retain tractability, we associate the (unobserved) resource landscape \$S(x)\$ with the current equilibrium population. At equilibrium, we assume \$P \propto S\$, and deviations from this equilibrium are penalized using the \$\chi^2\$-divergence \$D\_{\chi^2}(P^{\mathrm{CE}} ,\Vert, P\_+)\$, as simplified in Eq. 2. The framework can accommodate the following deviations from equilibrium:
>
>
> * Small departures from equilibrium: The divergence penalty is continuous, so if the actual population distribution $P_0$ is only approximately at equilibrium, the difference in penalty $D_{\chi^{2}}(P^{\mathrm{CE}}\Vert P_{+})$ remains within $O(\|P_0 - P_{+}\|)$. As long as the regularization weight $\lambda_2$ is appropriately tuned, this discrepancy remains a second-order effect, and the resulting CCE solution stays near-optimal.
>
> * Known transient distribution: If the population distribution $P_t$ evolves over time for $t \in [0, T]$, and a reliable estimate of the (possibly non-equilibrium) distribution $P_T$ is available at the time an individual requests a recourse—e.g., obtained from recent data or a population dynamics model—then we can directly replace the penalty term as follows:
> $$
> D_{\chi^{2}}(P^{\mathrm{CE}} \Vert P_{+})
> \quad\longrightarrow\quad
> D_{\chi^{2}}(P^{\mathrm{CE}} \Vert P_T).
> $$
> The rest of the derivation remains unchanged. Proposition 2 guarantees existence and optimality under the sole condition that the reference distribution has full support, which still holds.
>
> * Fully dynamic settings: If $P_t$ evolves over time, we employ the path-guided extension from Section 4.1, which optimises an entire trajectory $T_t$ and penalises the time-integrated divergence:
> $$
> \int_0^1 D_{\chi^{2}}((T_t)_{\\#} P \Vert P_t) \, dt.
> $$
> This reformulation turns CCE into a dynamic optimal transport problem, solvable using efficient algorithms such as the back-and-forth method.
>
> * Unknown but estimable equilibrium: If neither $P_t$ nor $S(x)$ is known, we can fit the mean-field PDE described in Appendix C to historical data to estimate a future $\hat{P}_T$. We then use $D_{\chi^{2}}(P^{\mathrm{CE}} \Vert \hat{P}_T)$ as the penalty. So Our methods rely only on samples, so no closed-form expression for the density is required.
>
> In summary, CCE is robust to moderate deviations from equilibrium. Moreover, the framework is flexible: the reference distribution can be adapted, or the full dynamic formulation can be applied when the population is far from equilibrium. These extensions ensure that CCE remains broadly applicable in both static and evolving environments. We thank the reviwer for this question and will clarify this in our updated manuscript.
>
> > Re. citations
>
> We apologize for the inconsistent use of citation commands. We ensure the use of citep and citet is consistent in our update.

---

> > ### Author Response · Authors · 2025-08-05
> >
> > Dear Reviewers,
> >
> > We deeply appreciate your valuable time and comments on our submission. As we approach the end of the discussion period, we would be grateful if you could kindly share any additional feedback or clarifications based on our earlier rebuttal. Your insights are extremely valuable to us in improving our manuscript.
> >
> > Thank you again for your thoughtful consideration.

---

> > ### Comment · Reviewer_J14q · 2025-08-06
> >
> > I appreciate the authors' response. Unfortunately, I think the core concerns I raised were not adequately addressed. Rather than responding directly to the formatting issue I pointed out, the authors shifted the discussion toward the clarity of their writing and motivation, which was not the focus of my critique.
> >
> > My concerns regarding the structure of the paper are as follows:
> > - Why is there a placeholder on page 8? This appears to **violate NeurIPS formatting requirements**.
> > - Why does the final page leave at least five lines unused? Taken together with the placeholder on page 8, this raises concerns that the submission may represent incomplete work.
> > - The paper lacks a conclusion. I disagree with the authors’ claim that a “discussion and future work” section suffices as a conclusion. This does not align with conventions in academic writing.
> >
> > Additionally, as Reviewer *kVUx* pointed out, the validity and significance of the experimental results remain questionable. The authors did not provide reproducibility materials at submission time. Instead, they linked to a GitHub repository from a previous work, which is highly unusual. Furthermore, **the authors’ attempt to direct reviewers to an external GitHub link during the rebuttal contradicts point 3 of this year’s NeurIPS rebuttal guidelines. This is unfair to other submissions that strictly followed the rules.**
> >
> > Given these concerns, I believe the paper requires a fresh round of review to determine whether it is suitable for acceptance.

---

> > > ### Author Response · Authors · 2025-08-06
> > >
> > > > Rather than responding directly to the formatting issue I pointed out, the authors shifted the discussion toward the clarity of their writing
> > >
> > > We respectfully note that the reviewer assigned a **clarity score of 1 to our submission**, a rating we found deeply uncalibrated. **We felt compelled to address this in our rebuttal**, as clarity is a central component of the paper’s evaluation.
> > >
> > > > Why is there a placeholder on page 8? This appears to violate NeurIPS formatting requirements.
> > >
> > > **There is no placeholder on page 8**. We believe the reviewer is referring to the stray period at the end of the figure caption, which was inadvertently introduced due to a LaTeX syntax issue. Here is our source LaTeX for this figure, which shows how this dot has been mistakenly placed after } (we have fixed it in our revision):
> > >
> > > ```
> > > \begin{figure}
> > >     \centering
> > >     \includegraphics[width=0.45\textwidth]{figs/compate.pdf}
> > >     \includegraphics[width=0.45\textwidth]{figs/adult_cce_10.pdf}
> > >     \caption{(Left) The blue curve represents the percentage increase in modification cost of CCE relative to standard CE as $\lambda_2$ varies from 0.01 to 0.3. The red curve illustrates the competition cost obtained by $\lambda_2 D_{\chi^2}(T_{\#}\bP_\ms \parallel \bP_\ps)$. Both curves are supported with confidence intervals. As expected, there is a trade-off between modification and competition cost measures. (Right) The result of CCE on the Adult dataset with $\lambda_2 = 0.1$.}.
> > >     \label{fig:tradeoff}
> > > \end{figure}
> > > ```
> > >
> > > Even after this fix, the vertical spacing between the figure and the surrounding text remains, a behavior dictated by the NeurIPS style file and entirely outside our control. **We reject any suggestion that our submission violates NeurIPS formatting requirements**. On the contrary, we have fully adhered to the provided formatting guidelines causing this space.
> > >
> > > > Why does the final page leave at least five lines unused?
> > >
> > > We respectfully note that at most four additional lines could fit on page 9. **Our manuscript is already at 30 pages, and we are in fact constrained for space. We would have gladly used the remaining lines if it were possible to do so meaningfully.**
> > >
> > > > Taken together with the placeholder on page 8, this raises concerns that the submission may represent incomplete work.
> > >
> > > Even if the reviewer remains unconvinced by our clarification that no placeholders were used, we find it concerning that the completeness of our submission is being questioned based on four unused lines at the bottom of the final page. We respectfully but firmly disagree with any assessment of scientific completeness that hinges on such formatting minutiae, rather than the substance of the work.
> > >
> > > > I disagree with the authors’ claim that a discussion and future work section suffices as a conclusion
> > >
> > > As noted above, we were constrained by the page limit and chose to devote the final page to additional related work, discussion, and future directions, which we believed provided a meaningful and well-rounded closure to the paper. That said, we respect the reviewer’s assessment and are happy to add a concise conclusion section in our revision.
> > >
> > > > The authors did not provide reproducibility materials at submission time. ... the authors’ attempt to direct reviewers to an external GitHub link during the rebuttal contradicts point 3 of this year’s guidelines.
> > >
> > > We respectfully disagree with the reviewer’s characterization. **In the submitted version, we provided complete details of our algorithms and hyperparameters**. We also cited a public GitHub repository used for baseline implementations, as is standard when building on prior work. The reason we did not share our own repository at submission time was solely to preserve anonymity.
> > >
> > > During the rebuttal phase, in response to the reviewer’s request, we created an anonymous GitHub repo with our implementation. To remain within the rebuttal guidelines, we did not include a direct link in our response. Instead, we made the code publicly available under a clearly identifiable name so that reviewers interested in verifying our implementation could do so at their discretion. We believe this was a good-faith effort to address the reviewer’s concern while complying with the guidelines.
> > >
> > > > I believe the paper requires a fresh round of review
> > >
> > > We are deeply disappointed that a minor formatting issue (4 unused lines on the final page) and our good-faith effort to ensure transparency are cited as reasons for requiring another round of review. We respectfully disagree with this assessment. **We believe the current manuscript is complete and fully suitable for evaluation as is. It includes detailed descriptions of our algorithms and experimental setup, along with a comprehensive appendix covering all relevant preliminaries and technical details. Minor formatting artifacts, especially those that would require only trivial revisions, should not impede an assessment of the scientific merit of our work.**

---

> > > > ### Comment · Area_Chair_YSww · 2025-08-06
> > > > **Dear Reviewer J14q and Authors**
> > > >
> > > > **Dear Reviewer J14q**
> > > >
> > > > I would advise you to focus on the contents of the paper rather than the formatting issues.
> > > > Such issues can be easily fixed at the time of submission, and it is not appropriate to reject the paper solely by this reason.
> > > > Please be constructive even if you are not positive towards this paper.
> > > >
> > > > **Dear Authors**
> > > >
> > > > Regarding the Github link, the claim of Reviewer J14q is appropriate.
> > > > It is not mandatory to the reviewers to check the external link during the reviewing periods.
> > > >
> > > > Thank you,
> > > >
> > > > AC

---

### Note · Authors · 2025-08-12

We thank the AC and reviewers for their thoughtful engagement. This work addresses a timely gap: **how to quantify and mitigate negative externalities in algorithmic recourse**. Standard counterfactual explanations optimize only individual modification costs without considering potential distributional shifts or increased competition when many act on similar recommendations. Our method avoids such artificial setups and makes the problem more realistic by accounting for the underlying distribution and generating collective recommendations.

Formally, our framework **Collective Counterfactual Explanations (CCE)** models population dynamics via mean-field game theory and penalizes deviations from equilibrium, thereby balancing individual costs with their externalities. This reframes CE as a **collective optimization problem** connected to optimal transport with existence and consistency guarantees as well as efficient algorithms with amortized inference. We also extend CCE to path-guided recourse and ordered classifier families.

During the discussion, we added experiments against four additional baselines and a runtime analysis. These results reinforced our main claim. We also clarified the robustness of our framework to the equilibrium assumption: CCE remains stable under moderate departures from equilibrium, can (i) adapt to known transient reference distributions, and (ii) use the dynamic, path-guided extension to handle evolving populations. The method operates directly on samples and remains compatible with black-box models.

Reproducibility is our priority. The submission includes pseudocode (Alg 1 & 4), full hyperparameters, open-source baselines, and detailed experimental protocols. We initially omitted the code link to preserve anonymity; at the reviewer’s request, we have now published an anonymous repository and referred reviewers to search our paper title on GitHub.

We will correct the minor formatting artifacts noted by Reviewer J14q and add a concise conclusion section.

**In sum, CCE provides a principled, scalable, and extensible approach to socially aware recourse. We are glad reviewers found our approach novel (kVUx, Aeab), theoretically rigorous (J14q, kVUx, Aeab), experimentally supported (Aeab, fXWP), and addressing a well-motivated, realistic recourse problem (Aeab, fXWP), while being clear, well-written (kVUx, Aeab, fXWP), and complete (kVUx). We thank the reviewers and AC for their time, constructive feedback, and consideration.**

---

### Decision · Program_Chairs · 2025-09-17

**Decision:**

Accept (poster)

**Comment:**

This paper proposes a framework for collective counterfactual explanations, which generates group-wise counterfactuals while preserving the distribution of both perturbed and positive instances and minimizing modification costs.
The approach addresses resource competition issues that arise when counterfactuals cluster in similar favorable regions.
The reviewers appreciated the clear motivations, illustrative examples, and a novel characterization of resources.
There was a few comments on the clarity of the paper, this could be easily fixed at the time of publication.